# A systematic review and meta-analysis of physical exercise non-adherence and its determinants among type 2 diabetic patients in Ethiopia

**Hailemicahel Kindie Abate**[1]*, **Abere Woretaw Azage**[1], **Alebachew Ferede Zegeye**[1], **Sintayehu Sime Tsega**[1], **Muluken Chanie Agimas**[2], **Habtamu Sewunet Mekonnen**[1], **Gashaw Adane Nega**[3], **Zarko Wako Beko**[1], **Chilot Kassa Mekonnen**[1]

1 Department Medical Nursing, College of Medicine and Health Science, University of Gondar, Gondar, Ethiopia, 2 Institute of Public Health Department of Epidemiology and Biostatistics, College of Medicine and Health Sciences, University of Gondar, Gondar, Ethiopia, 3 Department of Immunology and Molecular Biology, School of Biomedical and Laboratory Science, College of Medicine and Health Science, University of Gondar, Gondar, Ethiopia

* haile206k@gmail.com

**Editor:** Philipp Baumert, UMIT TIROL Private University for Health Sciences and Technology GmbH: UMIT TIROL Private Universitat fur Gesundheitswissenschaften und -technologie GmbH, GERMANY

## Abstract

### Introduction

Physical exercise non-adherence is one of the leading risk behavioral factors for type two diabetic patients and one of the leading causes of mortality of patients worldwide. Therefore, the current study was conducted to determine the pooled prevalence and its determinants of non-adherence to physical exercise among type two diabetes adult patients in Ethiopia.

### Methods

Studies were searched systematically using International databases from PubMed, Google Scholar, Cochrane Library, Embase, and CINAHL. The quality of articles that were searched was assessed using the New Castle Ottawa scale for a cross-sectional study design. Statistical analysis was performed using STATA version 14 and a meta-analysis was carried out using a random effect model method. Assessment of the certainty evidence's was done by applying the GRADE method. The Preferred Reporting Item for Systematic Review and Meta-analyses (PRISMA) guideline was followed for reporting results. The title and the protocol of this meta-analysis were registered at the online database PROSPERO registration number CRD42023430579.

### Result

From the total 1711 records screened, 7 studies with 3437 participants who fulfilled the inclusion criteria were included in this systematic review. The estimated pooled prevalence of exercise non-adherence in Ethiopia was 50.59%. Being female (OR = 1.27, 95% CI (1.82, 1.97)), primary level education (OR = 1.19, 95% CI (1.01, 1.39)) and rural residency (OR = 4.87, 95% CI (2.80, 8.48)) were significantly associated with exercise non-adherence.

**Data Availability Statement:** All relevant data are within the manuscript and its Supporting Information files.

**Funding:** The author(s) received no specific funding for this work.

**Competing interests:** The authors have declared that no competing interests exist.

**Abbreviations:** CI, confidence interval; COVID-19, coronavirus disease 2019; IDF, International Diabetes Federation; OR, odds ratio; PRISMA, preferred reporting items for systematic review meta-analyses guidelines; STATA, software for statistics and data analysis.

## Conclusion

According to papers evaluated by the GRADE assessment the certainty of evidence's was poor. More than half of the diabetes patients had physical exercise non-adherence. Strategies such as emotional support, health education, and emphasis on rural diabetic patients can improve the problem of non-adherence.

## Introduction

Physical exercise is regular body movement performed to improve or maintain individual health and fitness. Across the globe, the prevalence of chronic diseases such as diabetes mellitus is increasing with changes in lifestyle including physical inactivity [1]. The findings from the International Diabetic Federation (IDF) estimate that more than 451 million diabetes by the year 2017. This figure will rise to 693 million diabetes cases by the year 2045 [2].

Non-adherence to physical exercise is one of the leading risk behavioral factors among patients with chronic disease and one of the leading causes of disease globally. Nearly 30% of diabetic patients were thought to be caused by due to physical exercise inactivity [3].

Evidence from Bangladesh showed that about 25% of type 2 diabetes mellitus patients were non-adherence to exercise [4]. A Study conducted in China showed that low mean score of exercise adherence among type 2 diabetes mellitus patients [5]. Another evidence from Nepal showed that 42.1% of the study participants with type 2 diabetes were non-adherent to regular physical exercise [6]. In South Africa, the prevalence of non-adherence to physical exercise in diabetes mellitus patients was estimated at 50% [7]. Regular Physical exercise is vital to decrease the burden of chronic disease and decrease the consequence of other psychological problems such as depression and anxiety disorders [8, 9]. In recent years, evidence has shown that physical inactivity is the mainstay management component of diabetic populations [10, 11]. Although physical exercise has so many benefits, it remains undermined in the self-care management of diabetic patients. This problem is frequently observed in the developing diabetic population due to a sedentary lifestyle [12].

Nowadays, physical inactivity is one of the serious public health impacts with a significant impact on type 2 diabetic patients [13]. According to the 2010 World Health Organization report, physical inactivity is one of the leading causes of mortality with an estimated 3.2 million deaths occurring due to physical inactivity, which is an easily preventable chronic disease [8].

Physical exercise adherence has a multidimensional importance in improving the impact of COVID-19 and other comorbid chronic diseases including diabetes population [14]. Physical exercise non-adherence has to be improved by implementing high-intensity training exercises and mobilization of the community to physical exercise [15–17].

Evidence showed that physical exercise non-adherence was determined by many factors such as sex of participants, education and residency, economic status, partner participation, and individual perception were among the commonest significant factors of physical exercise non-adherence of diabetes patients [7, 18, 19].

In the past decades, solutions have been tried to implement the challenges of non-adherence to physical exercise in the diabetic population. Giving health education on different occasions and preparing documents such as leaflets that are easily accessible by the clients about the importance of physical exercise on the improvement of diabetes and other comorbid diseases were among the attempted solutions [20, 21].

Different studies have been conducted in Ethiopia to determine physical exercise non-adherence and its determinants among type 2 diabetic patients. The prevalence of physical

exercise non-adherence had a high prevalence (92%) and low (11.9%) in studies conducted in the Amhara and Oromia region states of Ethiopia, respectively. The findings of these different studies showed that there was a significant discrepancy in the prevalence of non-adherence to physical exercise among type 2 diabetes adult patients in Ethiopia.

As far as the search of the researchers, there was no systematic review and meta-analysis conducted in Ethiopia related to this review. The findings of this study will be used as feedback to healthcare administrators to design or revise policies on the implementation of non-adherence to physical exercise among type 2 diabetes. Therefore, this study was conducted to determine the pooled prevalence and its determinants of non-adherence to physical exercise among type 2 diabetes in Ethiopia.

### Research question

What is the prevalence of physical exercise non-adherence and its determinants among type 2 diabetic patients in Ethiopia?

Is there an association between physical exercise non-adherence and being a type 2 diabetic patient in Ethiopia?

## Methods

The current systematic review and meta-analysis utilized the guideline of the Newcastle-Ottawa Scale for cross-sectional studies for systematic review and meta-analysis [22, 23], and the report is written consistent with the revised 2020 PRISMA checklist [24] (S1 File). This systematic review and meta-analysis title and its protocol were registered in the PROSPERO online database registration number CRD42023430579.

### Searching strategies

The indexed traditional database (Medline/PubMed, Scopus, Web of Science, Embase, and Cochrane Library) and grey literature (Google, OpenGrey, Google Scholar, and ProQuest) were used to locate research articles on the prevalence of physical exercise non-adherence type two diabetic patients in Ethiopia. The string for searching was developed using "AND" and "OR" Boolean operators with the keywords extracted from the Medical Subject Headings (MeSH) database. The search strategy was based on the research question of this review and utilized CoCoPop (Co = Condition, Co = Context, Pop = Population) for prevalence and PEO (Pop = Population, E = exposure, O = outcome interest) for determinant factors.

Using PubMed/Medline the article locating strategy was through (((((((((physical exercise non-adherence) OR (physical inactivity)) OR (exercise none-adherence)) OR (exercise adherence)) OR (prevalence physical exercise non-adherence)) AND (type two diabetes)) OR (diabetes)) OR (insulin resistance diabetes)) AND (Ethiopia). In the same way, the Embase Emtree, Cochrane library, Scopus OpenGrey, and ProQuest were used as a database based on the modeled search strategy design for Embase Cochrane and Scopus using title (Ti) and abstract (Ab) (S2 File). This search strategy primarily aimed to trace all reviewed (published) and unpublished primary studies. The list of all retrieved primary articles, systematic reviews, and meta-analyses references were also screened or cross-referenced to get extra studies. The sources of information range from electronic databases to direct contact with the principal investigator if mandatory. The first search through PubMed, Cochrane Library, Scopus, Web of Science, Google, and Google Scholar was done in April 2023. The final search for updating was conducted from June 5/ 2023 to June 29/ 6/2023. The publication date was used as a filter mechanism in which articles published from January 1/ 2013 to June 29/2023 included the current systematic review and meta-analysis study to generate the most recent evidence for the

scientific community. To access articles, there were no year or language restrictions in the database search.

### Eligibility criteria

**Study inclusion and exclusion criteria.**   Quantitative studies that reported the prevalence of overall physical exercise non-adherence of type 2 diabetic patients, master's thesis, and dissertations were included in the study, whereas qualitative study design, single case study research reports, and not fully accessed articles were excluded from the analysis.

### Study selection and outcome

After comprehensive searching, all located citations were selected and exported to Endnote Citation Manager Software version X7. Following this, irrelevant and duplicated articles were removed. Then two independent researchers (CKM and AWA) screened each particular article for its title, abstract, and full text by far and cross-checked it against the inclusion criteria. The other researcher team (HKA, SSC, MCA, and AFZ) checked the screened articles with full text for details under already defined criteria to take it to the final review process. Any sort of disagreement between the research team while including and excluding articles could be solved by discussion based on the predefined criteria. The result of searching further screening and inclusion process of articles in this review was done in agreement with the PRISMA guidelines for Systematic Review and Meta-analysis 2020.

### Quality appraisal of included studies

The modified version of the Newcastle-Ottawa Scale for cross-sectional studies quality assessment tool is used to evaluate the quality of the included studies [23]. This critical appraisal tool consists of 3 items which asses the selection of study subjects (which consists of 4 different questions which account for a maximum of 5 points), the comparability of the study (which has 2 different questions accounting for 3 points for each) and the outcome of the study (which contain 1 question and maximum 1 point for each question) (S1 Table). After the critical appraisal, the reviewers decided to include or exclude screened articles based on the overall quality of the appraisal score out of 9. The article was prone to exclude when the score was below average, which is of three independent reviewers.

Two independent reviewers assessed each paper's quality before inclusion in the review using 9 points from the three sections of the tool.

The quality of each study was rated using three categorical algorithms. A score $\geq 7$ was considered as "good quality" of the study, a score rated as 4–6 "fair quality" of studies, and studies with quality $\leq 3$ was considered as "poor quality". Studies with a final quality score of 4 points and above were included in the final review [23].

In this regard, there had to be 2 studies [25, 26] with poor quality categories for the article to be excluded from the review. This critical appraisal threshold was supported by previously published systematic review and meta-analysis studies [23, 27]. Any sort of disagreement between the involved reviewers was solved through the discussion of the reviewers. Furthermore, if the disagreement unfolds, the second reviewer was indicated to oversee the source of doubt and reach to consensus. The exclusion of the articles was presented with countable reasons which could be consistent with the low-quality score of the articles. The result of searching further screening and inclusion process of articles in this review was done in agreement with the PRISMA guidelines for Systematic Review and Meta-analysis 2020.

## Data extraction

Data were independently extracted by two authors using a standardized data extraction format of JBI as developed according to the 2014 Joanna Briggs Institute Reviewers' Manual [28]. The tools including authors, study year, study design, sample size, prevalence, and risk of bias assessment score were included in the extraction. The data were extracted by two independent reviewers (HKA and SST) and any inconsistent data was cross-checked (S2 Table). The disagreement between the reviewers was solved through by discussion. To handle missing data, we have used a standardized data extraction form for each single study included and also conducted a robust inspection of our data before formal analysis took over. In addition, we have conducted sensitivity and trim and fill analysis to detect and adjust potential publication bias (S3 File).

## Outcome measurements

Regarding this particular review, physical exercise non-adherence is defined as a patient who has not applied 30 minutes of regular exercise per day (like brisk walking, strength training, and stretching exercise) or 150-minute aerobic physical exercise for at least 5 days to keep the blood glucose level in the normal state [21, 29].

   The primary outcomes of this meta-analysis and systematic review were the pooled prevalence of exercise non-adherence among type 2 diabetes patients elsewhere in Ethiopia.

## Data synthesis and analysis

The outcome of the included primary studies was narratively presented and expanded with supplementary materials in text, tables, and figures where necessary. All necessary and relevant information from each article was extracted through a Microsoft Excel spreadsheet and exported to STATA Version 11 for further analysis. The random-effects model was employed to estimate the pooled effect size of exercise non-adherence among type 2 diabetes patients due to the presence of heterogeneity [30]. Heterogeneity or variation between reported prevalence was evaluated using $I^2$ and Cochran's Q statistics of 25%, 50%, and 75% with mild, moderate, and high heterogeneity with a p-value less than 0.05 [31]. The pooled effect size with a 95% confidence interval was presented in the forest plot and used to visualize the presence of heterogeneity graphically. For the possible difference in the primary study, we explored subgroup analysis and meta-regression subsequently using publication year, study design, study setting, sampling methods, sample size, sex, and region. Moreover, the publication bias was assessed through visual inspection of the funnel plot, Begg- Mazumdar Rank correlation test, and Egger's test to see the funnel plot's asymmetry [32]. The influence of individual articles on the overall pooled effect size estimate or the prevalence of exercise non-adherence among type 2 diabetes was assessed by using sensitivity analysis. The forest plot with 95% CI was used to present the overall pooled prevalence as well as the subgroup pooled prevalence of exercise non-adherence among type 2 in Ethiopia. The log odds ratio was used to determine the associated factors of physical exercise non-adherence among type 2 diabetes patients in Ethiopia.

   **Certainty assessment.**   The selected meta-analysis articles were assessed concerning the quality of evidence scored in the five domains specified within GRADE (Grade of Recommendations, Assessment, Development, and Evaluation) on the limitation including risk bias, inconsistency of results, indirectness of evidence, impression of results, and publication bias [33]. The overall grade rating results are based on four levels (high, moderate, low, and very low). The supplementary file (S3 Table) shows a more detailed description of the GRADE process. The GRADE judgment for the reviews of the articles was assessed independently of the findings of the original review authors. The first author (HKA) and the second author (AWZ)

assessed and verified the rating of the final included articles and disagreements were settled after discussion.

## Result

### Systematic review

**Article selection and outcome.** In this systematic review and meta-analysis study, a total of 1914 articles related to the prevalence of exercise non-adherence in Ethiopia were identified using electronic databases and search engine websites. Among the overall articles found 1201 were removed for being irrelevant and duplicated and the other 231 and 113 were removed for not being eligible (study design and title difference) by automation tools and other reasons, respectively. The remaining 88 articles were eligible for screening (S4 Table). Of these screened, 51 papers were excluded due to the region of study or not being conducted in Ethiopia and target population differences (those articles were conducted among type one diabetes). On further screening, 23 articles were sought for retrieval and 8 were not retrieved for one and the other reason. Moreover, 14 research articles were assessed for eligibility to be included in the review process, but with the outcome of interest and measurement tool ambiguity, a total of 7 articles were excluded. Finally, 7 original research articles were incorporated into this systematic review and meta-analysis (Fig 1).

**Description of included studies.** All research articles included in this systematic review were done by cross-sectional study design and published from January 1/2013 to June 29/2023. Seven published studies with 3437 participants were included to determine the pooled prevalence of physical exercise non-adherence among type 2 diabetes patients. All articles were conducted with a cross-sectional study design with the smallest prevalence from Oromia (11.9%) [34] and the largest prevalence from Amhara 73.6% [35] regional states of Ethiopia. On the contrary, the largest sample size was from the Oromia (1191) [36], whereas the smallest sample size was from the Amhara (302) regional states of Ethiopia [37]. This review includes three studies from the Amhara region [35, 37, 38] and four studies from the Oromia regional states of Ethiopia [34, 36, 39, 40] (Table 1).

**Quality of included studies.** There are a total of 7 articles assessed for quality scores using the 9 points Newcastle-Ottawa Scale for cross-sectional studies quality assessment tool [23]. The outcome of the quality appraisal ranged from moderate to high-quality scores, in which two studies scored 9 points [36, 39], three studies scored 8 [34, 35, 38], and the other two studies scored 7 [37, 40]. Beside, two studies scored 3 based on the quality appraisal assessment. Therefore, it were excluded from the analysis (S3 File).

### Meta-analysis

**The pooled prevalence of physical exercise non-adherence.** The prevalence of physical exercise non-adherence ranges from 11.7% in the Oromia [34] region to 73.6% among type 2 diabetes patients in the Amhara [35] regional states of Ethiopia. The pooled prevalence of physical exercise non-adherence among type 2 diabetes patients was 50.59% with 95% CI (24.59–66.58) based on the DerSimonian-Laird random effect model analysis (Fig 2).

The lowest pooled prevalence of physical exercise non-adherence of type 2 diabetes patients was found in the Oromia region at 43.85% (95% CI 18.66–69.04) and the highest pooled prevalence in Amhara regional state of Ethiopia at 59.57% (95% CI (35.03–84.10) (Fig 3).

**Assessment of heterogeneity.** In this systematic review, subgroup analysis was done to assess the heterogeneity, so the p-value and $I^2$ statistics were used to assess the heterogeneity between studies. The analysis result showed that the source of heterogeneity could be due to the region (p = 0.0001, $I^{2} = 100$) (Fig 2)

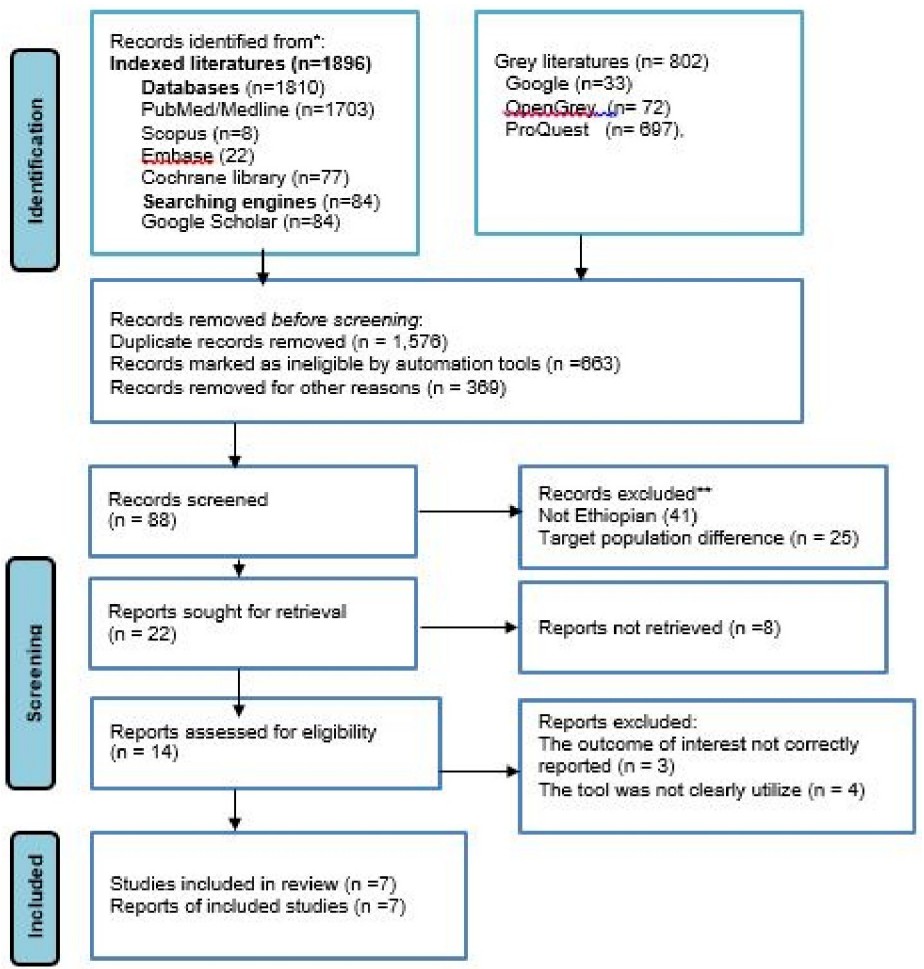

**Fig 1. PRISMA flow diagram of the included studies.**

The source of heterogeneity was further assessed using study year, sampling techniques, or study settings to identify the reason for variation among studies, but none of them are the source of heterogeneity (Table 2).

**Publication bias.** The funnel plot was done to show to check the publication bias. This study has no publication bias since it is symmetrically distributed. Furthermore, statistically,

**Table 1. Summary of the prevalence of physical exercise non-adherence among seven studies included in the systematic review and meta-analysis.**

| Author/year of publication | | Region | Sample method | Sample size | Outcome | Prevalence | Response rate (%) | Quality |
|---|---|---|---|---|---|---|---|---|
| Debalke et al. [39] | 2022 | Oromia | systematic random sampling | 392 | 243 | 38 | 92.9 | 9 |
| Zenu el. al. [36] | 2023 | Oromia | mult-stage sampling | 1191 | 729 | 61.2 | 93.3 | 9 |
| Abate et al. [35] | 2020 | Amhara | systematic random sampling | 576 | 450 | 73.6 | 99.3 | 8 |
| Enyew et. Al. [37] | 2023 | Amhara | systematic random sampling | 302 | 93 | 72 | 98 | 7 |
| Edmealem Et.al. [38] | 2020 | Amhara | systematic random sampling | 332 | 110 | 33.1 | 91.2 | 8 |
| Negra et.al [40] | 2020 | oromia | simple random sampling | 322 | 206 | 64.3 | 100 | 6 |
| Tamirat et.al [34] | 2014 | oromia | systematic random sampling | 322 | 41 | 11.9 | 99.1 | 8 |

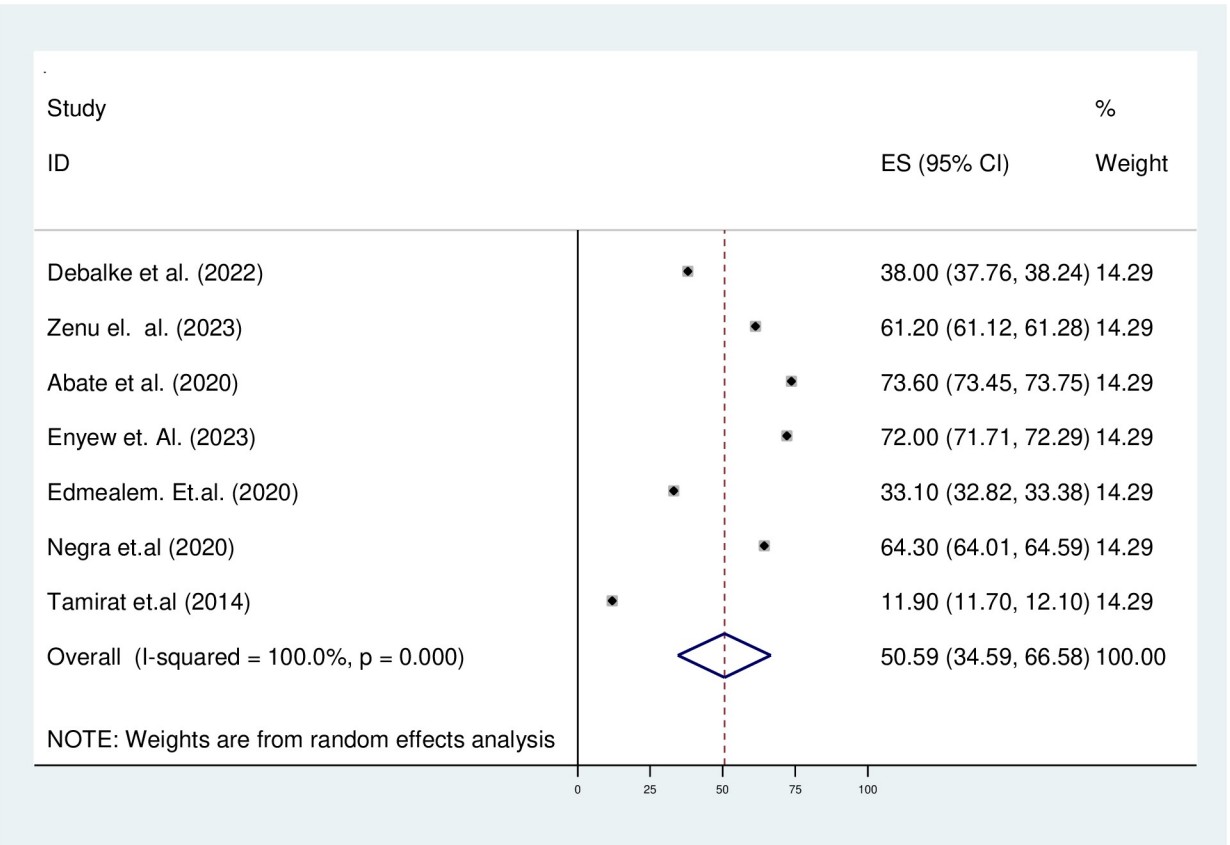

**Fig 2. Forest plot of the prevalence of physical exercise non-adherence among type 2 diabetes patients in Ethiopia.**

Begg's test and Egger test were done with p-value = 0.86 which showed that there was no publication bias (Fig 4).

**Sensitivity analysis.** A sensitivity analysis was performed after observing lower and higher values of the review to show the effect of one study on the overall pulled summary effect. However, the analysis result of the sensitivity test using the random-effects model indicated that no single affected the overall estimate (Fig 5).

**The pooled analysis of determinant factors.** *The effect of the sex of participants.* Being female was 1.27 times increase in the overall pooled prevalence of physical exercise non-adherence among type 2 diabetes patients (OR = 1.27, 95% CI (1.82, 1.97)) (Fig 6).

**Heterogeneity and publication bias of the included studies.** As stated in Fig 6, the overall heterogeneity test ($I^2$) on the effect of female sex was 0.0% with a p-value of 0.977, using a random effect model to adjust the observed variability. This heterogeneity test indicates there is no observed variability across the included studies. Regarding the publication bias, the funnel plot showed that symmetrical graphical presentation which indicates no publication bias (Fig 7). Furthermore, statistically, the Egger test was done with p = 0.31 which showed that there was no publication bias.

**The effect of primary education.** Primary education was a contributing factor for physical exercise non-adherence with 1.19 times increase in the overall pooled prevalence of physical exercise non-adherence among type 2 diabetes patients (OR = 1.19, 95% CI (1.01, 1.39)) (Fig 8).

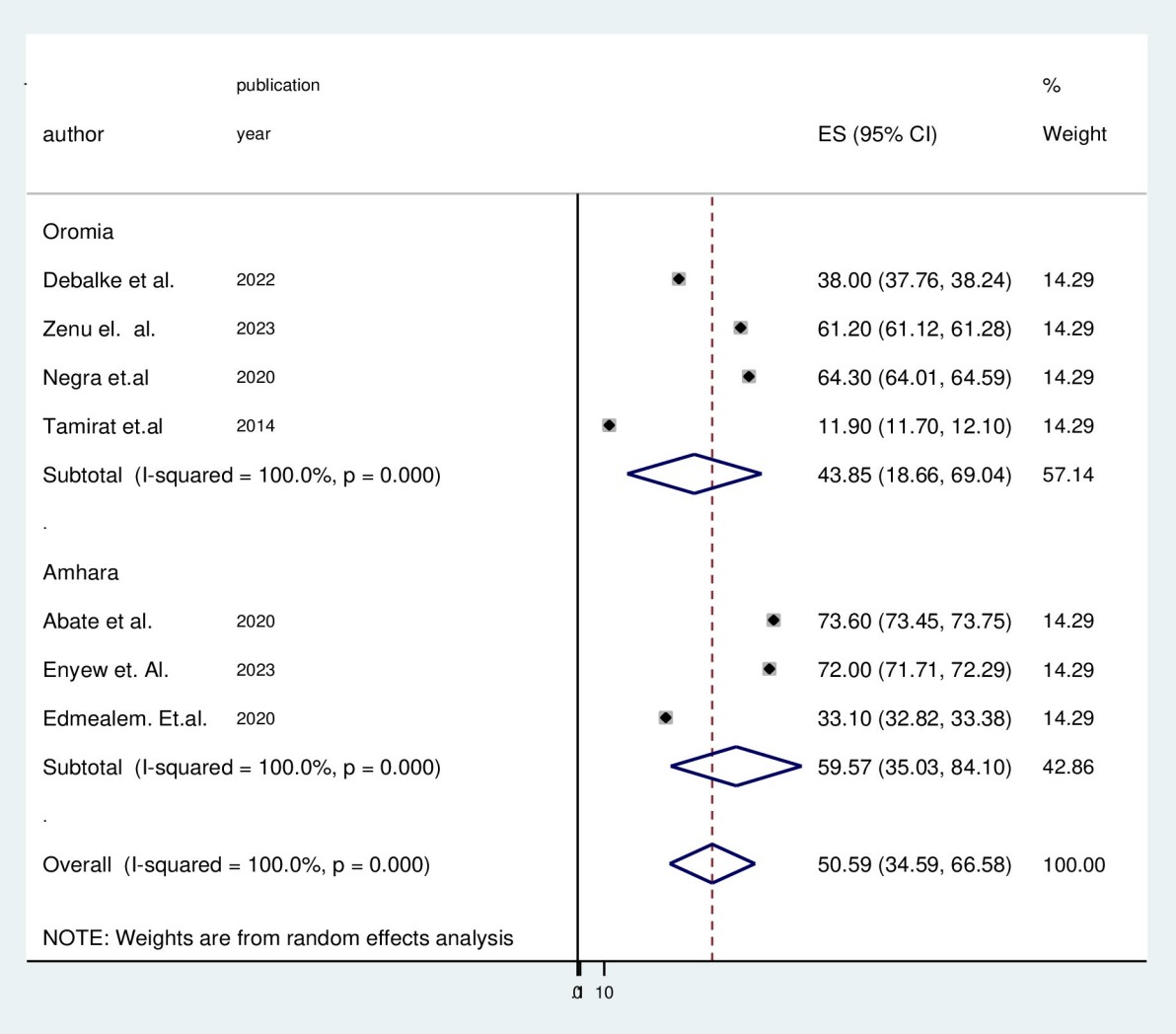

**Fig 3. The forest plot of subgroup analysis by region of 7 included studies.**

**Heterogeneity and publication bias of the included studies.** As stated above in Fig 8, the overall heterogeneity test ($I^2$) on the effect of primary education was 0.0% with a p-value of 0.548, using a random effect model to adjust the observed variability. This heterogeneity test

**Table 2. Subgroup analysis of the pooled prevalence of physical exercise non-adherence among type 2 diabetes patients in Ethiopia.**

| subgroups | | number of studies | Pooled prevalence | heterogeneity statistics | p-value | $I^2$ | Tau squared |
|---|---|---|---|---|---|---|---|
| Sampling technique | Multistage | 2 | 67.40(55.24–79.55) | 150000 | < 0.001 | 100% | 76. 87 |
| | Systematic random sampling | 5 | 43.86 (21.63–66.08) | 20418 | < 0.001 | 100% | 642.69 |
| Study period | < 2020 | 3 | 51.47 (36.67–66.27) | 150000 | < 0.001 | 100% | 1300 |
| | ≥ 2020 | 4 | 50.58 (34.59–66.57) | 99948.91 | < 0.001 | 100% | 227.98 |
| Study setting | Hospital | 5 | 43.86(21.64–66.08) | 150000 | P<001 | 100% | 642.69 |
| | Community | 2 | 67.40(55.25–79.55) | 20418.30 | P<001 | 100% | 76.87 |

NB: *others; systematic and stratified sampling

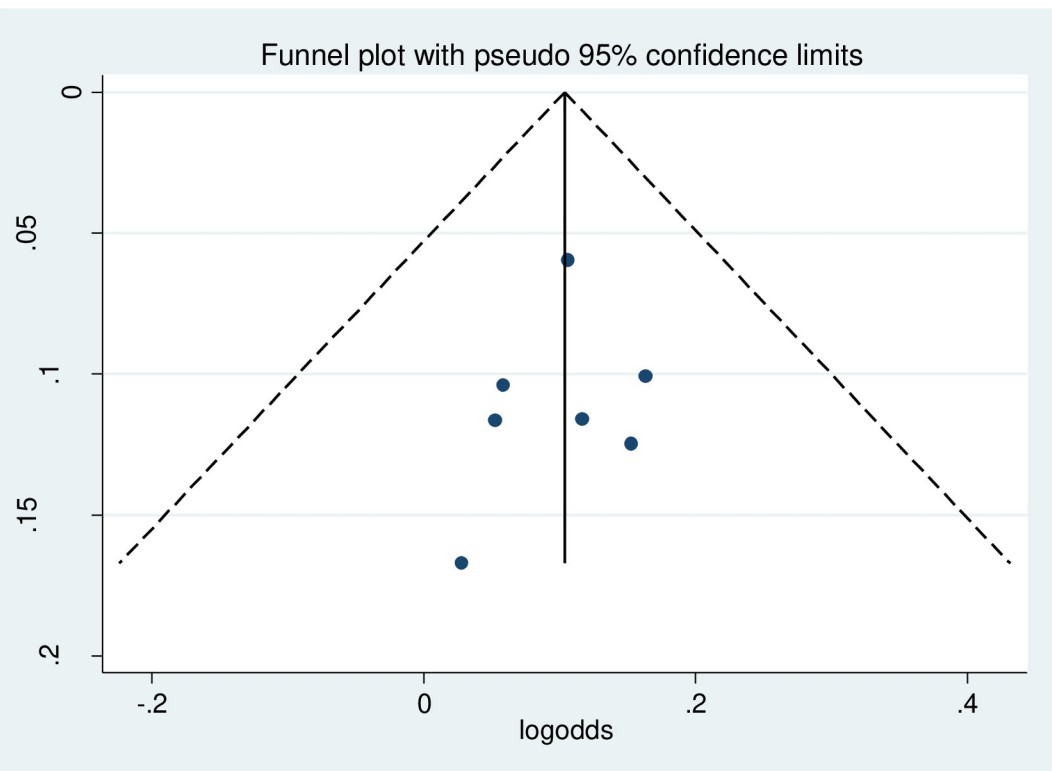

**Fig 4. Funnel plot to of physical exercise non-adherence among type 2 diabetes in Ethiopia.**

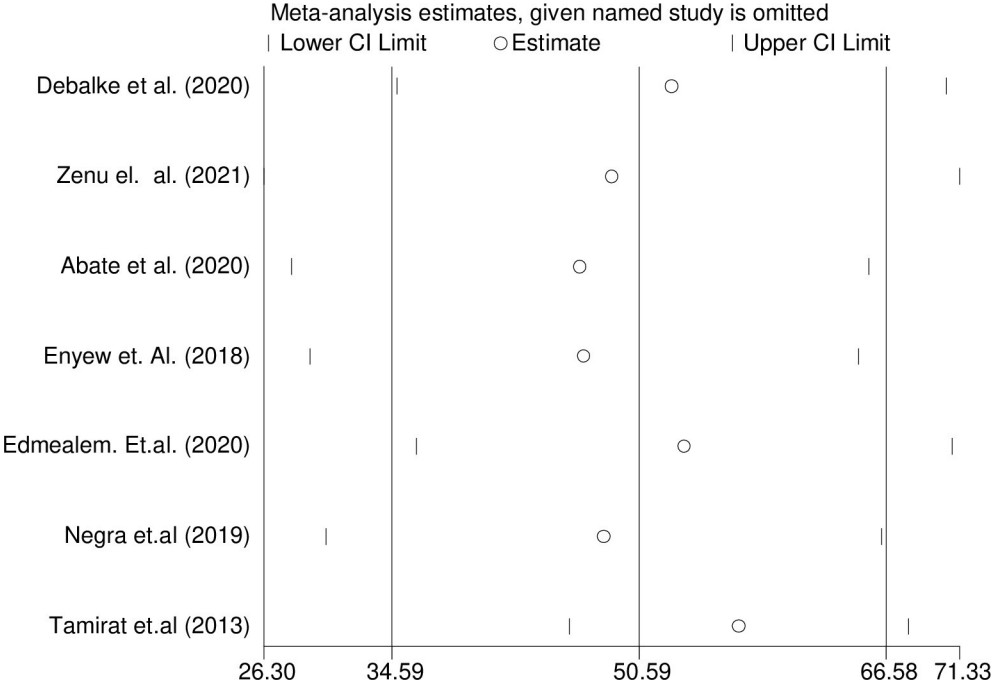

**Fig 5.** Sensitivity analysis which indicates the physical exercise non-adherence among type 2 diabetes in Ethiopia.

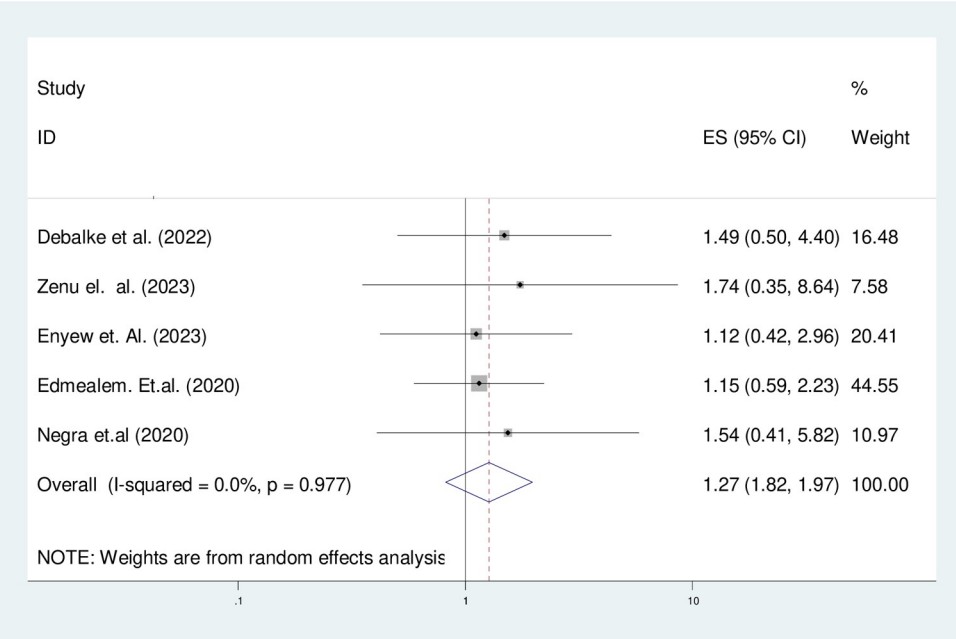

**Fig 6. The forest plot of asses pooled effect of sex of participants on physical exercise non-adherence on type 2 diabetic patients in Ethiopia.**

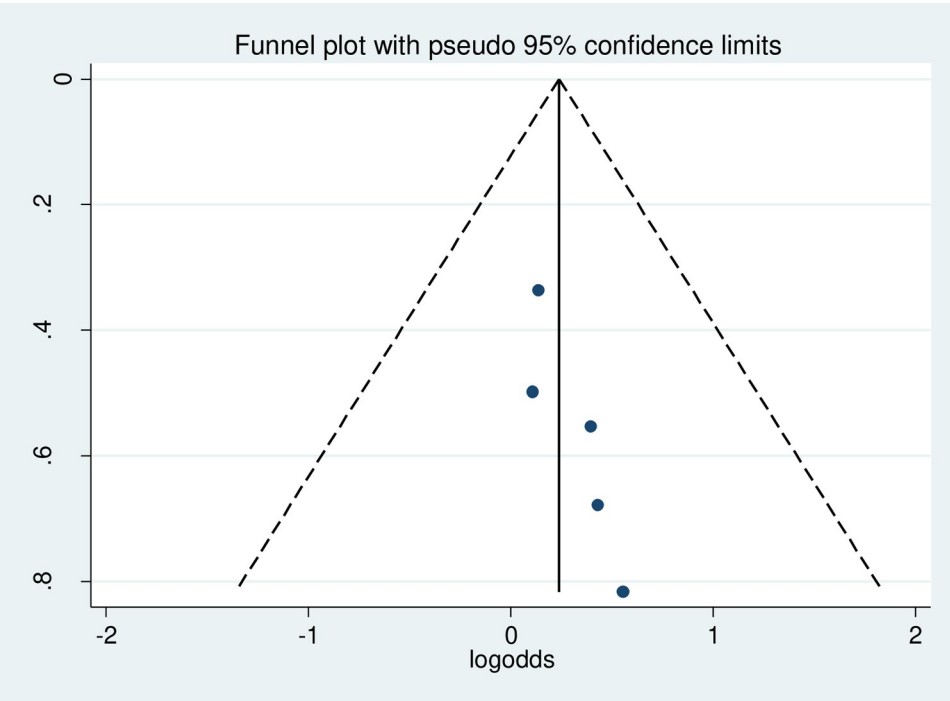

**Fig 7. Funnel plot to assess the publication bias on the effect of sex of participants on physical exercise non-adherence among type 2 diabetes in Ethiopia.**

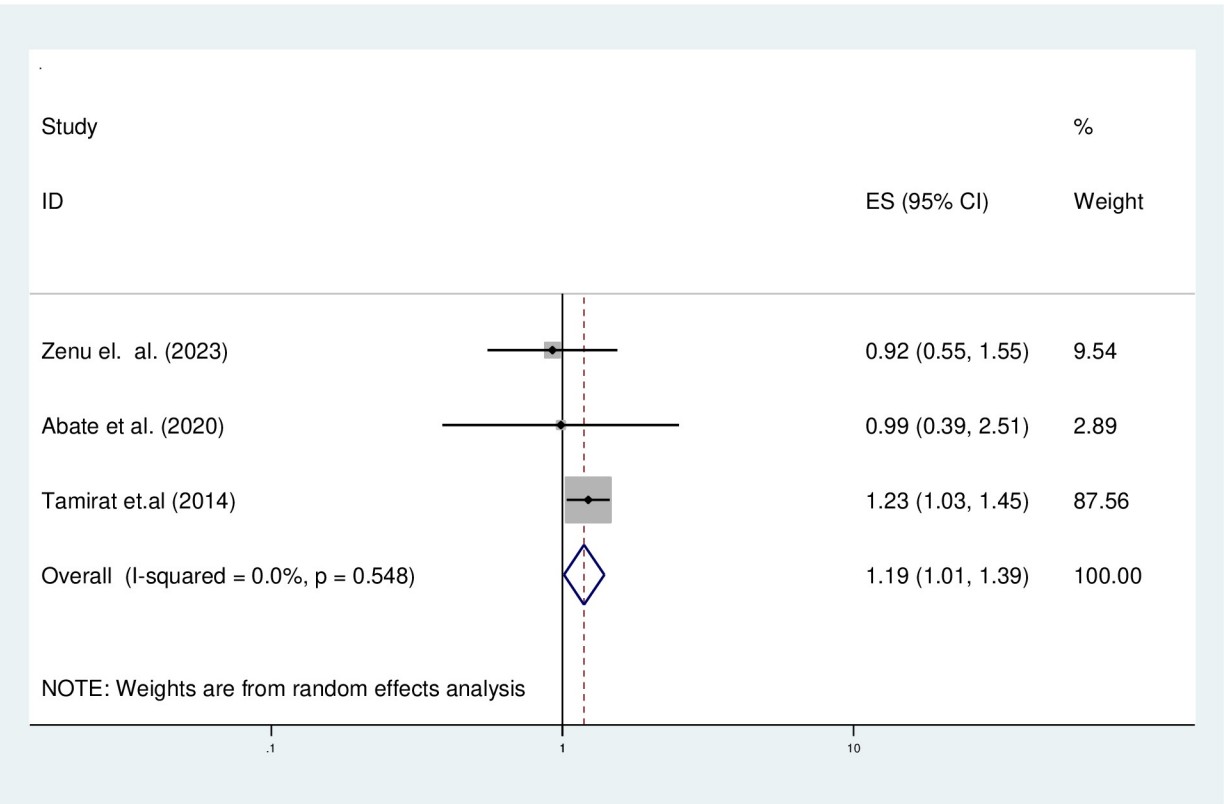

**Fig 8. The forest plot of asses pooled effect of primary education of participants on physical exercise non-adherence on type 2 diabetic patients in Ethiopia.**

indicates there was no observed variability across the included studies. Regarding the publication bias, the funnel plot showed that symmetrical graphical presentation which indicates no publication bias (Fig 9). Furthermore, statistically, the Egger test was done with p = 0.31 which showed that there was no publication bias.

**The effect of rural residency.** Being rural residency was a contributing factor for physical exercise non-adherence with a 4.87 times increase in the overall pooled prevalence of physical exercise non-adherence among type 2 diabetes patients (OR = 4.87, 95% CI (2.80, 8.48)) (Fig 10).

**Heterogeneity and publication bias of the included studies.** As stated above in Fig 10, the overall heterogeneity test ($I^2$) on the effect of being in rural residency was 20.2% with a p-value of 0.057, using a random effect model to adjust the observed variability. This heterogeneity test indicates there was mild observed variability across the included studies. Regarding the publication bias, the funnel plot showed that symmetrical graphical presentation which indicates no publication bias (Fig 11). Furthermore, statistically, the Egger test was done with p = 0.134 which showed that there was no publication bias.

**Assessment of certainty of evidence.** Since all of the included studies were cross-sectional, our assessment of the evidence's certainty was poor. Moreover, papers evaluated the evidence's certainty by applying the GRADE certainty evaluation criteria's three upgrading and five downgrade aspects [33]. The ranking indicates that the articles lack consistency because not all of the studies were included in the review. The included evidence does not exhibit publication bias, as demonstrated by the Egger test and funnel plot. Because the sample size of the evidence was large and the confidence interval of the pooled prevalence was small,

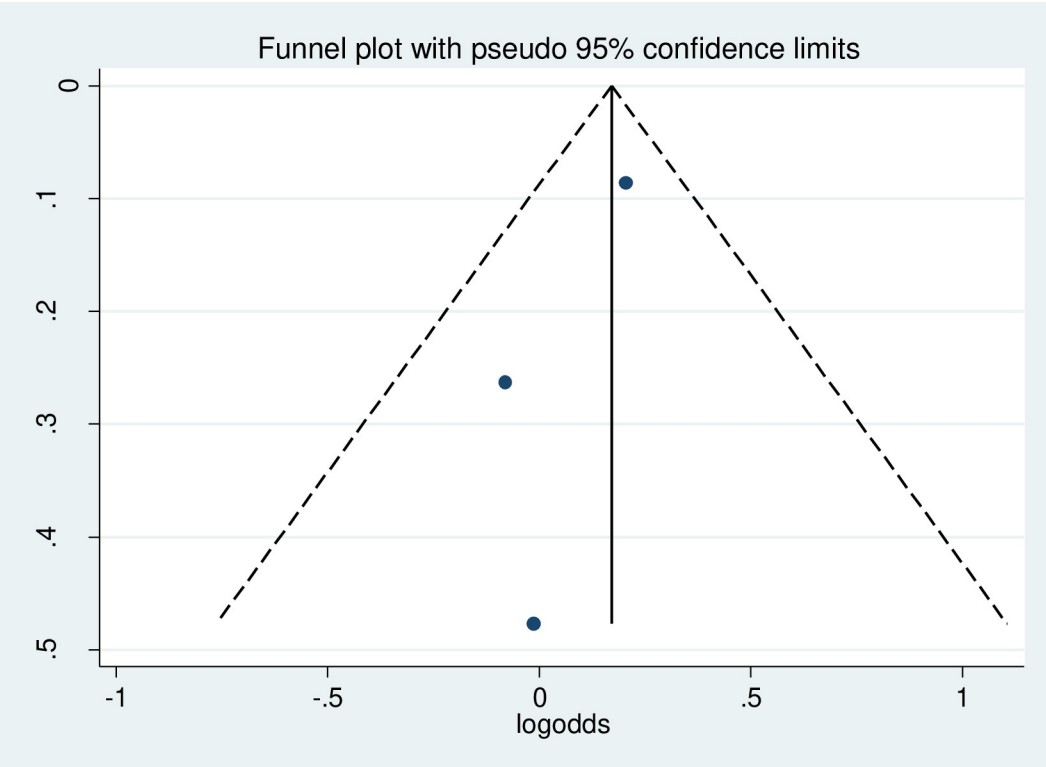

**Fig 9. Funnel plot to assess the publication bias on the effect of primary education of participants on physical exercise non-adherence among type 2 diabetes in Ethiopia.**

there was no conclusive evidence of imprecise measurement of the outcome variable articles. This systematic and meta-analysis's outcome variable showed no indirectness. Furthermore, every included study that examined the outcome variable did so directly; none of the included studies measured the outcome variable indirectly. The estimates of the variable adjusted for the confounding variables, and in all included studies, the outcome variable's estimate was sufficiently large. It is generally expected that the existing rate of physical exercise non-adherence among people with type diabetes in Ethiopia will alter if further studies are included; therefore, more research is necessary to provide more precise evidence that we can use to make decisions (S3 Table).

## Discussion

Physical exercise is one of the mainstay management of diabetic patients. This lifestyle management is particularly important for those with type 2 diabetes [41]. Poor adherence to physical exercise recommendations leads to poor glycemic control and chronic diabetic complications [42]. This study attempted to investigate the pooled prevalence of physical exercise non-adherence and its determinants among type 2 diabetes patients in Ethiopia.

In this systematic review and meta-analysis, the pooled prevalence of physical exercise non-adherence among diabetes patients was 50.59% with 95% CI (24.59–66.58). This finding was in line with the study conducted in Botswana (52%) [7]. This figure showed that more than half of the type 2 diabetes participants were non-adherence to physical exercise. The main possible justification might be due to a lack of education/ information about the importance of physical exercise adherence. Evidence also supports that exercise non-adherence matters due

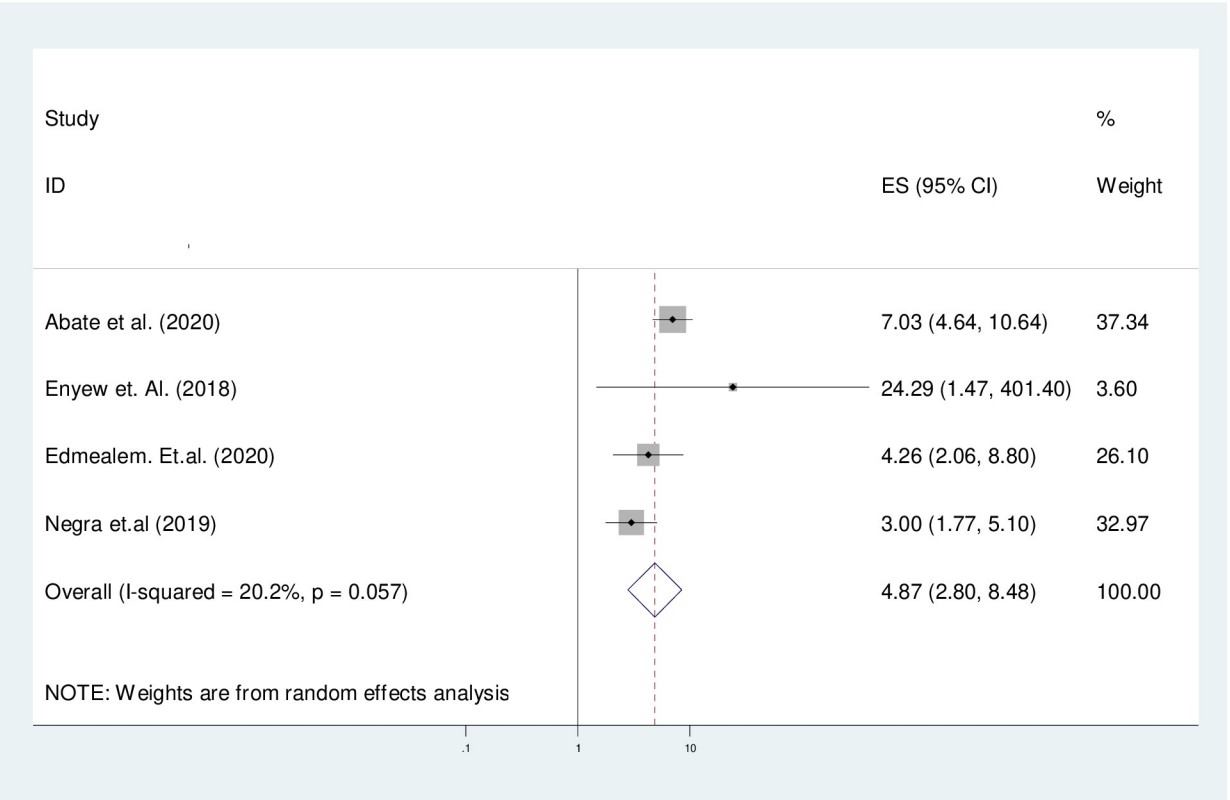

**Fig 10. The forest plot of asses pooled effect of rural residency of participants on physical exercise non-adherence on type 2 diabetic patients in Ethiopia.**

to the wrong perception that exercise can aggravate the prognosis of diabetes [43]. This finding is lower than the study conducted in West Africa (13%) [44]. The possible justification might be due to the study conducted in West Africa including many countries compared to the current study, which might lead to a decrease in the pooled prevalence of physical inactivity. The finding is also supported by the studies conducted in the USA [45, 46], and Australia [47, 48].

In this meta-analysis, the pooled determinants such as being female sex, having primary education, and being rural residency were significantly associated with physical exercise non-adherence among type 2 diabetes patients. Therefore, being female was a 1.27 times increase in the overall pooled prevalence of physical exercise non-adherence among type 2 diabetes patients compared to male participants. Even though there was no similar study, the finding was supported by the study conducted in Botswana [7]. This might be due to a lack of exercise partners, lack of motivation, and lack of community support for females [43].

Primary education was a 1.19 times increase in the overall pooled prevalence of physical exercise non-adherence among type 2 diabetes patients. This finding was supported by a study conducted in Switzerland [49]. This is because education can increase a person's information about the importance of lifestyle modification, including physical exercise [50].

Rural residency was 4.87 times increase in the overall pooled prevalence of physical exercise non-adherence among type 2 diabetes patients compared to urban residency. This finding was supported by the study conducted in Pakistan [51]. Although rural diabetes is physically more active than participants in urban areas, they might have no regular schedule

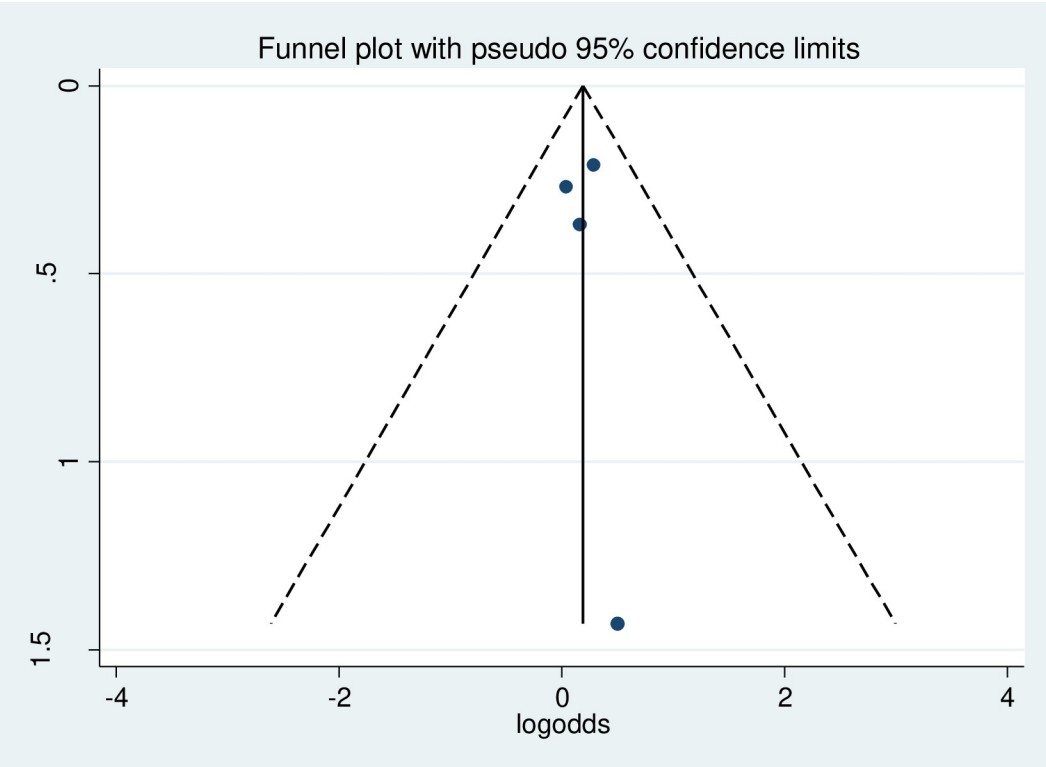

**Fig 11. Funnel plot to assess the publication bias on the effect of rural residency of participants on physical exercise non-adherence among type 2 diabetes in Ethiopia.**

for physical exercise due to many reasons such as health information access about physical exercise non-adherence [52].

This study had its own limitations. In this study, qualitative studies were not included so it might be more informative if both qualitative and quantitative methods had been incorporated. Studies included in this review were cross-sectional as a result; the outcome variable might be affected by other confounding variables, and the GRADE overall quality score becomes low. Moreover, some regions were not incorporated because a lack of research may lead to an underestimate of this review.

## Conclusions and recommendations

In this meta-analysis, all included studies assessed for evidence's certainty, and its overall quality was poor based on GEADE criteria. The detail was described on the supplementary file attached as S3 File and S3 Table. This systematic review and meta-analysis revealed that more than half of type two diabetic patients had physical exercise non-adherence compared to most of the previously conducted studies. In this studies, variables such as being female sex, having primary education, and being rural residency were significantly associated with physical exercise non-adherence among type 2 diabetes patients. This s national evidence would be helpful for cross-country comparisons of the proportion of exercise non-adherence among type 2 diabetes patients. It might be useful for healthcare policymakers to emphasize the overall quality of service by incorporating components of lifestyle modification, particularly physical exercise adherence for diabetes patients. Improving strategies such as empowering women, preparing leaflets about diabetic information, and organizing health education about the scientific merit

of exercise for diabetic patients are among the best modalities to improve the problem of exercise non-adherence. In addition, special attention should be given to rural diabetic patients since they have poor exercise adherence compared to urban diabetic patients.

## Supporting information

**S1 File. PRISMA checklist.**
(DOCX)

**S2 File. Search strategy.**
(DOCX)

**S3 File. Handling missing data.**
(DOCX)

**S1 Table. Quality appraisal of included studies.**
(DOCX)

**S2 Table. Extracted data from primary research source.**
(DOCX)

**S3 Table. GRADE assessment of selected meta-analysis studies.**
(DOCX)

**S4 Table. All recorded articles in the search.**
(DOCX)

## Acknowledgments

We would like to thank all authors included in this systematic review and meta-analysis.

## Author Contributions

**Conceptualization:** Hailemicahel Kindie Abate, Abere Woretaw Azage, Alebachew Ferede Zegeye, Sintayehu Sime Tsega, Muluken Chanie Agimas, Habtamu Sewunet Mekonnen, Gashaw Adane Nega, Zarko Wako Beko, Chilot Kassa Mekonnen.

**Data curation:** Hailemicahel Kindie Abate, Abere Woretaw Azage, Alebachew Ferede Zegeye, Sintayehu Sime Tsega, Muluken Chanie Agimas, Habtamu Sewunet Mekonnen, Gashaw Adane Nega, Zarko Wako Beko, Chilot Kassa Mekonnen.

**Formal analysis:** Hailemicahel Kindie Abate, Abere Woretaw Azage, Alebachew Ferede Zegeye, Sintayehu Sime Tsega, Muluken Chanie Agimas, Habtamu Sewunet Mekonnen, Gashaw Adane Nega, Zarko Wako Beko, Chilot Kassa Mekonnen.

**Funding acquisition:** Hailemicahel Kindie Abate, Abere Woretaw Azage, Alebachew Ferede Zegeye, Sintayehu Sime Tsega, Muluken Chanie Agimas, Habtamu Sewunet Mekonnen, Gashaw Adane Nega, Zarko Wako Beko, Chilot Kassa Mekonnen.

**Investigation:** Hailemicahel Kindie Abate, Abere Woretaw Azage, Alebachew Ferede Zegeye, Sintayehu Sime Tsega, Muluken Chanie Agimas, Habtamu Sewunet Mekonnen, Gashaw Adane Nega, Zarko Wako Beko, Chilot Kassa Mekonnen.

**Methodology:** Hailemicahel Kindie Abate, Abere Woretaw Azage, Alebachew Ferede Zegeye, Sintayehu Sime Tsega, Muluken Chanie Agimas, Habtamu Sewunet Mekonnen, Gashaw Adane Nega, Zarko Wako Beko, Chilot Kassa Mekonnen.

**Project administration:** Hailemicahel Kindie Abate, Abere Woretaw Azage, Alebachew Ferede Zegeye, Sintayehu Sime Tsega, Muluken Chanie Agimas, Habtamu Sewunet Mekonnen, Gashaw Adane Nega, Zarko Wako Beko, Chilot Kassa Mekonnen.

**Resources:** Hailemicahel Kindie Abate, Abere Woretaw Azage, Alebachew Ferede Zegeye, Sintayehu Sime Tsega, Muluken Chanie Agimas, Habtamu Sewunet Mekonnen, Gashaw Adane Nega, Zarko Wako Beko, Chilot Kassa Mekonnen.

**Software:** Hailemicahel Kindie Abate, Abere Woretaw Azage, Alebachew Ferede Zegeye, Sintayehu Sime Tsega, Muluken Chanie Agimas, Habtamu Sewunet Mekonnen, Gashaw Adane Nega, Zarko Wako Beko, Chilot Kassa Mekonnen.

**Supervision:** Hailemicahel Kindie Abate, Abere Woretaw Azage, Alebachew Ferede Zegeye, Sintayehu Sime Tsega, Muluken Chanie Agimas, Habtamu Sewunet Mekonnen, Gashaw Adane Nega, Zarko Wako Beko, Chilot Kassa Mekonnen.

**Validation:** Hailemicahel Kindie Abate, Abere Woretaw Azage, Alebachew Ferede Zegeye, Sintayehu Sime Tsega, Muluken Chanie Agimas, Habtamu Sewunet Mekonnen, Gashaw Adane Nega, Zarko Wako Beko, Chilot Kassa Mekonnen.

**Visualization:** Hailemicahel Kindie Abate, Abere Woretaw Azage, Alebachew Ferede Zegeye, Sintayehu Sime Tsega, Muluken Chanie Agimas, Habtamu Sewunet Mekonnen, Gashaw Adane Nega, Zarko Wako Beko, Chilot Kassa Mekonnen.

**Writing – original draft:** Hailemicahel Kindie Abate, Abere Woretaw Azage, Alebachew Ferede Zegeye, Sintayehu Sime Tsega, Muluken Chanie Agimas, Habtamu Sewunet Mekonnen, Gashaw Adane Nega, Zarko Wako Beko, Chilot Kassa Mekonnen.

**Writing – review & editing:** Hailemicahel Kindie Abate, Abere Woretaw Azage, Alebachew Ferede Zegeye, Sintayehu Sime Tsega, Muluken Chanie Agimas, Habtamu Sewunet Mekonnen, Gashaw Adane Nega, Zarko Wako Beko, Chilot Kassa Mekonnen.

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
