## [Decision Letter · Decision Letter 0]

26 Feb 2024

PONE-D-23-20714A systematic review and meta-analysis of physical exercise non-adherence and its determinants among type 2 diabetic patients in Ethiopia:PLOS ONE

Dear Dr. Abate,

Thank you for submitting your manuscript to PLOS ONE. After careful consideration, we feel that it has merit but does not fully meet PLOS ONE’s publication criteria as it currently stands. Therefore, we invite you to submit a revised version of the manuscript that addresses the points raised during the review process.

**Dear Authors,****please see the comments made by the reviewers. There are some crucial points that need to be improve as recommended for systematic review and meta-analysis manuscripts.**** **==============================

We look forward to receiving your revised manuscript.

Kind regards,

Thiago Gomes Heck, Ph.D.

Academic Editor

PLOS ONE

Journal Requirements:

- https://www.sciencedirect.com/science/article/pii/S2214139122000142?via%3Dihub

- https://www.dovepress.com/getfile.php?fileID=68447

In your revision ensure you cite all your sources (including your own works), and quote or rephrase any duplicated text outside the methods section. Further consideration is dependent on these concerns being addressed.

6. Please include a separate caption for each figure in your manuscript.

7. Please include a copy of Table 1 and 2 which you refer to in your text on page 11.

Additional Editor Comments:

The manuscript is interesting and can found specialized audience. However, some fragilities in description and interpretation of the results need to be repair before acceptance.

Please, see the carefull comments made by the reviewers to improve some parts of the manuscript.

Reviewers' comments:

Reviewer's Responses to Questions

**Comments to the Author**

1. Is the manuscript technically sound, and do the data support the conclusions?

Reviewer #1: Yes

Reviewer #2: Yes

Reviewer #3: Partly

2. Has the statistical analysis been performed appropriately and rigorously? 

Reviewer #1: Yes

Reviewer #2: Yes

Reviewer #3: No

3. Have the authors made all data underlying the findings in their manuscript fully available?

Reviewer #1: Yes

Reviewer #2: Yes

Reviewer #3: No

4. Is the manuscript presented in an intelligible fashion and written in standard English?

Reviewer #1: No

Reviewer #2: Yes

Reviewer #3: Yes

5. Review Comments to the Author

Reviewer #1: The article written in a comprehensive manner and justifies the title. However, we came across several errors that needs to be corrected. In abstract, please specify the type of diabetic in introduction first line “type diabetic patients”. In methods, replace the word systematic review to meta-analysis in the sentence “Statistical analysis was performed using STATA version 14 and a systemic review was carried out using a random effect model method”.

Correct the date format - June 5/ 2023 to June 29/ 6/2023 and January 2013 to June 29/2023. Complete the sentence “Any sort of disagreement between the research team while including and excluding articles on predefined criteria”

Please specify, data was extracted by two reviewers or four reviewers “Data were independently extracted by four authors using a standardized data extraction format of JBI as developed according to the 2014 Joanna Briggs Institute Reviewers’ Manual [26]. The tool includes Authors, study year, study design, sample size, prevalence, and risk of bias assessment score were included in the extraction. The data were extracted by two independent reviewers and any inconsistent data was cross-checked (supplementary file: Table S2).”

Please correct 113 articles are not being ineligible or not being eligible “the other 231 and 113 were removed for not being ineligible (study design and title difference) by automation tools and other reasons respectively.”

Correct the spelling of “among typ1 diabetes”, “asses”, “suplimentary table” and “metedological data quality”

Please recheck the number of excluded studies 74-51 is 23 not 22. Methodological quality assessment would come after the description of studies. Maintain the consistency of terminology throughout the article.

Correct the sentence in conclusion “as compared to most of the previous studies conducted else were in the world”. Author’s contribution should be in proper format. Correct the referencing format in reference no. 4 and 5.

Overall, major revisions are recommended.

Reviewer #2: The present study is a systematic review and meta-analysis on non-adherence to physical activity in patients diagnosed with type 2 diabetes mellitus in the Ethiopian population. The structure of the study is very well thought out in terms of conduction, data extraction and selection of included studies. A very positive point for the quality of the study was that the statistical tools for conducting meta-analysis, meta-regression and sensitivity analysis were present in the manuscript. The written language is adequate, although it presents some small spelling errors that do not significantly affect the overall quality of the manuscript. Despite these positive points and which demonstrate that the researchers were concerned with conducting the study in the best way possible, some pillars of the manuscript's structure worry me, as they may have failed to include studies in this systematic review. Please look carefully at my suggestions/questions/comments below:

1. Do the authors have the project for this systematic review registered on the PROSPERO platform? The CRD approval number must be present on the abstract and in the methods section.

2. Please complete the checklist for the abstract in accordance with the PRISMA 2020 protocol and include it as a supplementary document to the manuscript.

3. The authors used EMBASE as a database, but only used terms indexed for MeSH. I strongly recommend that authors carry out a new search with terms indexed from the European platform that is specifically directed to EMBASE, Emtree.

4. Why did the authors not include gray literature as an additional strategy? It is described in the methodology very vaguely, but the selected articles are not present in Figure 1, whether they were preprints or even the gray literature bases used for this systematic review. For this systematic review design, I believe it is an interesting addition to broaden the range of researchers.

5. Please add the search strategy with the terms MeSH, Emtree with their indexed terms and their synonyms as a supplementary document. In this case, I suggest including search terms for observational studies (as was the focus of this manuscript). I even suggest that authors learn more about the studies used in this Cochrane review published in 2019, which evaluated precisely this issue for search engines.

Reference:

Li L, Smith HE, Atun R, Tudor Car L. Search strategies to identify observational studies in MEDLINE and Embase. Cochrane Database of Systematic Reviews 2019, Issue 3. Art. No.: MR000041. DOI: 10.1002/14651858.MR000041.pub2. Accessed 05 January 2024.

https://www.cochranelibrary.com/cdsr/doi/10.1002/14651858.MR000041.pub2/full

6. I suggest that the authors carry out a new search with more diversified terms in the domain of adherence to physical exercise, as the terms used by the authors may bring fewer studies than the databases can have (my concern is related to the indexing of terms for the primary outcome established by the authors for this systematic review). To do this, I suggest reading supplementary Table 1 of the article cited below as inspiration:

Reference:

El Haddad L, Peiris CL, Taylor NF, McLean S. Determinants of Non-Adherence to Exercise or Physical Activity in People with Metabolic Syndrome: A Mixed Methods Review. Patient Prefer Adherence. 2023 Feb 3;17:311-329. doi: 10.2147/PPA.S383482. PMID: 36760232; PMCID: PMC9904214.

Reviewer #3: Dear Editors and Authors.

Thank you for giving me the opportunity to get to know your work and to collaborate with you.

Please find below what I have contributed.

The study aims to assess the prevalence and reasons for non-adherence to physical activity in people with DM2.

The introduction is appropriate but, although justified, it seems too focused on the country in question. This may limit the external validity of the study.

In the methodology, the authors clearly describe the research strategies, the database, the tools used to assess the risk of bias in the studies, and the statistical methods used to analyze the data.

The presentation of the results is partially satisfactory (forest plot) and the discussion is in line with the use of the available data. The conclusion is partially consistent with the objectives of the study and the data found.

However, in my opinion, there are some points that need clarification, which, if considered by the authors, could contribute to the quality of the work.

Introduction

The introduction presents the research question well. However, although it is justified, it seems to me to be extremely local.

Methods

Inform PROSPERO registration number.

Searching strategies

Change “PubMed” to “Medline/PubMed”

Explain the reason for limiting the date range of the search. I see no reason to limit the search to the last 10 years.

Was grey literature searched? If so, where?

Eligibility criteria

Study Inclusion and Exclusion criteria.

Poor methodological quality is not a criterion for excluding studies. These studies must be included and analyzed. If necessary, a subgroup analysis can be performed.

I suggest including in the methods the reason for including in the subgroup analysis only the variables of sex, place of residence and level of education.

I recommend using GRADE to assess the quality of evidence generated by meta-analysis.

Results

Page 10, line 5 - …”those articles conducted among typ1 diabetes” * please, correct “type”.

I suggest that a table similar to Table 1 in the PRISMA checklist be included in the paper. In addition to the data in this table, I propose to include the variables analyzed, sex, place of residence and level of education.

Assessment of heterogeneity – page 11

“The analysis result showed that the source of heterogeneity is not due to region (p=0.0001 I2 =100).” I don’t understand this result. p<0,05 and I2 100% isn’t an indicator of heterogeneity? If I have understood this result correctly, it means that there is a high degree of heterogeneity (The authors indicated this condition when describing the analysis of heterogeneity in the methods). Furthermore, the difference between the states/regions, suggests this difference.

The pooled analysis of determinant factors - Page 12

The effect of the sex of participants

Heterogeneity and Publication bias of the included studies

“As stated figure 6 showed that the overall heterogeneity test (I2) on the effect of being female sex was 0.0% with a p-value of 0.977, using a random effect model to adjust the observed variability. This heterogeneity test indicates there is observed variability across the included studies.”

Again, in my opinion, the authors have confused the results of the heterogeneity test.

The effect of primary education

Heterogeneity and Publication bias of the included studies – page 13

“As stated above in Figure 8, the overall heterogeneity test (I2) on the effect of primary education was 0.0% with a p-value of 0.548, using a random effect model to adjust the observed variability. This heterogeneity test indicates there is observed variability across the included studies.”

Same comment as above.

The effect of rural residency

Heterogeneity and Publication bias of the included studies

“As stated above in Figure 10, the overall heterogeneity test (I2) on the effect of being in rural residency was 20.2% with a p-value of 0.057, using a random effect model to adjust the observed variability. This heterogeneity test indicates there is observed variability across the included studies.”

Same comment as above.

Discussion

My suggestion is to review the results of the heterogeneity analysis and, if my analysis is correct, to explore the reasons for the heterogeneity in the discussion, as it is not possible to do a meta-regression due to the small number of studies included.

In addition, I suggest that the individual quality of the studies and the quality of the evidence (GRADE) be examined in each of the topics of discussion.

Regarding the limitations of the study, I suggest highlighting the small number of studies and states included, and, depending on the quality of the evidence, reporting this as a limitation of the study.

Conclusion

In the conclusion of the study, my suggestion is to minimize the impact of the results due to the limitations that are presented.

6. PLOS authors have the option to publish the peer review history of their article (what does this mean?). If published, this will include your full peer review and any attached files.

Reviewer #1: No

Reviewer #2: **Yes: **Giuseppe Potrick Stefani

Reviewer #3: **Yes: **LUIS FERNANDO DERESZ

---

## [Author Response · Author response to Decision Letter 0]

6 Apr 2024

Authors’ point-by-point Response letter to the editor and Reviewers’ Reports

A systematic review and meta-analysis of physical exercise non-adherence and its determinants among type 2 diabetic patients in Ethiopia. 

Authors:

 Hailemicahel Kindie Abate1*, Abere Woretaw Azage1, Muluken Chanie Agimas2, Gashaw Adane Nega3, Chilot Kassa Mekonnen1

Authors’ response to Editor’s and Reviewers’ reports 

In the first place, we authors would like to give our heartfelt gratitude to Journal of “plos one” Editorial Team and the respective reviewers for reviewing our revised manuscript and providing the necessary comments that going to be corrected accordingly. As per the editors and reviewers’ comments, the authors tried to make corrections accordingly. This is a point-by-point letter to editors, reviewer 1, 2, and 3 reports. 

Note: the indicated page and line numbers are referred to the clean version of the manuscript 

PONE-D-23-20714

A systematic review and meta-analysis of physical exercise non-adherence and its determinants among type 2 diabetic patients in Ethiopia:

PLOS ONE

Dear Dr. Abate,

Thank you for submitting your manuscript to PLOS ONE. After careful consideration, we feel that it has merit but does not fully meet PLOS ONE’s publication criteria as it currently stands. Therefore, we invite you to submit a revised version of the manuscript that addresses the points raised during the review process.

Dear Authors,

please see the comments made by the reviewers. There are some crucial points that need to be improve as recommended for systematic review and meta-analysis manuscripts.

Authors’ response: Thank you Editor, we had submitted the disproof letter responded to each letter on comments given below. 

Authors’ response: Thank you Editor, and we had submitted highlighted copy of original version one. 

Authors’ response: We had labeled and submitted accordingly. 

We look forward to receiving your revised manuscript.

Kind regards,

Thiago Gomes Heck, Ph.D.

Academic Editor

PLOS ONE

Journal Requirements:

Authors Response: dear editor thank you for you constructive we had prepared as per the journal requirement . 

Authors Response: Thank you we will make with reasonable request. 

- https://www.sciencedirect.com/science/article/pii/S2214139122000142?via%3Dihub

- https://www.dovepress.com/getfile.php?fileID=68447

In your revision ensure you cite all your sources (including your own works), and quote or rephrase any duplicated text outside the methods section. Further consideration is dependent on these concerns being addressed.

Authors Response: We had paraphrased. See : sections of other than the method. 

Authors Response: We have submitted the data as supplementary file: See: page 16 line… 393

Authors Response: thank you, we had made similar both in online and in the manuscript 

6. Please include a separate caption for each figure in your manuscript.

Authors Response: Thank you we had made a separate caption for each figure in the manuscript. See: page 17, line,401-420

7. Please include a copy of Table 1 and 2 which you refer to in your text on page 11.

Authors Response: Thank you editor we had made correction in as per your comment. We had copied the tables under the text. See: page 10-12, 

Authors Response: we had included the Supporting Information as per your comment. See page 17…. Line 421-426

Additional Editor Comments:

The manuscript is interesting and can found specialized audience. However, some fragilities in description and interpretation of the results need to be repair before acceptance.

Please, see the carefull comments made by the reviewers to improve some parts of the manuscript.

Authors Response: thank you, we try to modify the all points of the reviewers. 

Reviewers' comments:

Reviewer's Responses to Questions

Comments to the Author

1. Is the manuscript technically sound, and do the data support the conclusions?

Reviewer #1: Yes

Reviewer #2: Yes

Reviewer #3: Partly

Authors Response: Thank reviewer 1 and 2, and we try to addressed the reviewer 3 concern on the specific comment, given below. 

2. Has the statistical analysis been performed appropriately and rigorously?

Reviewer #1: Yes

Reviewer #2: Yes

Reviewer #3: No

Authors Response: Thank reviewer 1 and 2, and we try to addressed the reviewer 3 concern on the specific comment, given below. 

3. Have the authors made all data underlying the findings in their manuscript fully available?

Reviewer #1: Yes

Reviewer #2: Yes

Reviewer #3: No

Authors Response: Thank reviewer 1 and 2, and we try to addressed the reviewer 3 concern on the specific comment, given below. 

4. Is the manuscript presented in an intelligible fashion and written in standard English?

Reviewer #1: No

Reviewer #2: Yes

Reviewer #3: Yes

Authors Response: Thank reviewer 2 and 3, and we try to address the reviewer 1concern on the specific comment, given below. 

5. Review Comments to the Author

Authors Response: Thank you we had seen all the concerns of authors.

Reviewer #1: The article written in a comprehensive manner and justifies the title. However, we came across several errors that needs to be corrected. In abstract, please specify the type of diabetic in introduction first line “type diabetic patients”.

Authors Response: thank you we had mad correction as “type two diabetic patients” see page 2… line.. 28..

 In methods, replace the word systematic review to meta-analysis in the sentence “Statistical analysis was performed using STATA version 14 and a systemic review was carried out using a random effect model method”.

Authors Response: thank you we had mad correction as “meta-analysis” see page 2… line.. 35..

Correct the date format - June 5/ 2023 to June 29/ 6/2023 and January 2013 to June 29/2023. 

Authors Response: we had made correction to the format “January 1/ 2013 to June 29/2023” See: .page 6, line 135-138

Complete the sentence “Any sort of disagreement between the research team while including and excluding articles on predefined criteria”

Authors Response: Thank you we had made correction and complete the sentence as, “Any sort of disagreement between the research team while including and excluding articles could be solved by discussion based on the predefined criteria”. Page 7, line, 178-181

Please specify, data was extracted by two reviewers or four reviewers “Data were independently extracted by four authors using a standardized data extraction format of JBI as developed according to the 2014 Joanna Briggs Institute Reviewers’ Manual [26]. The tool includes Authors, study year, study design, sample size, prevalence, and risk of bias assessment score were included in the extraction. The data were extracted by two independent reviewers and any inconsistent data was cross-checked (supplementary file: Table S2).”

Authors Response: thank reviewers for the inconsistency the data were actually extracted by two reviewers. We had made correction on this typing error. See: page 5 line 177, 183. 

Please correct 113 articles are not being ineligible or not being eligible “the other 231 and 113 were removed for not being ineligible (study design and title difference) by automation tools and other reasons respectively.”

Authors Response: thank you we had make correction on the wording problem. 

See: page 9, line 222 

Correct the spelling of “among typ1 diabetes”, “asses”, “suplimentary table” and “metedological data quality”

Authors Response: thank you we had make correction on the spelling, and we had revised the supplementary files. See: page 9, line 235

Please recheck the number of excluded studies 74-51 is 23 not 22. 

Authors Response: thank you we had make correction on the numbering. 

See page: 9 line 226

Methodological quality assessment would come after the description of studies. 

Authors Response: thank you we had made as per your comment. 

Maintain the consistency of terminology throughout the article.

Correct the sentence in conclusion “as compared to most of the previous studies conducted else were in the world”. 

Authors Response: we had made correction on the sentence as “…. most of the previous conducted studies”.

Author’s contribution should be in proper format. 

Authors Response: Thank you reviewer we had made correct ion on it. 

Correct the referencing format in reference no. 4 and 5. 

Authors Response: thank you we had made correction the format of reference 4and 5we had made correction on it. 

Overall, major revisions are recommended.

Reviewer #2: The present study is a systematic review and meta-analysis on non-adherence to physical activity in patients diagnosed with type 2 diabetes mellitus in the Ethiopian population. The structure of the study is very well thought out in terms of conduction, data extraction and selection of included studies. A very positive point for the quality of the study was that the statistical tools for conducting meta-analysis, meta-regression and sensitivity analysis were present in the manuscript. The written language is adequate, although it presents some small spelling errors that do not significantly affect the overall quality of the manuscript. Despite these positive points and which demonstrate that the researchers were concerned with conducting the study in the best way possible, some pillars of the manuscript's structure worry me, as they may have failed to include studies in this systematic review. Please look carefully at my suggesti

---

## [Decision Letter · Decision Letter 1]

31 Jul 2024

PONE-D-23-20714R1A systematic review and meta-analysis of physical exercise non-adherence and its determinants among type 2 diabetic patients in Ethiopia:PLOS ONE

Dear Dr. Abate,

Thank you for submitting your manuscript to PLOS ONE. After careful consideration, we feel that it has merit but does not fully meet PLOS ONE’s publication criteria as it currently stands. Therefore, we invite you to submit a revised version of the manuscript that addresses the points raised during the review process.

We look forward to receiving your revised manuscript.

Kind regards,

Thiago Gomes Heck, Ph.D.

Academic Editor

PLOS ONE

Journal Requirements:

Additional Editor Comments:

Dear authors,

The manuscript is interestinlgy and can be accept after minor changes.

Please observe carefully the comments made by reviewer 2 below:

Dear Editor and Authors!

Thank you for the opportunity to review the paper again. In this analysis I can see the effort and progress of the work, but in my opinion, there are still adjustments to be made that could improve the quality of the study.

Abstract – Use the PRISMA for Abstracts Checklist – include PROSPERO registration number

Methods

Searching strategies

Line 118 – Distinguish in the text between searches in traditional databases and searches in gray literature. In addition, Figure 1 needs to be modified to show searches and results from indexed databases and grey literature separately.

In addition, there are other databases besides Google for searching gray literature, clinicaltrials.gov, International Clinical Trials Registry Platform (ICTRP), The European Union Clinical Trials, OpenGrey, ProQuest.

125 – Correct the word "Cochrane" wherever it appears.

130 – S2 file - Present the complete search strategy used in each included database in the Supplementary Appendix.

137 – Correct the word MedLine

149 – Correct the word PubMed

In the text, state that there were no year or language restrictions in the database search.

Eligibility criteria

Study Inclusion and Exclusion criteria

Lines 157 to 160

Quantitative studies that reported the prevalence of overall physical exercise non-adherence of type 2 diabetic patients, master's thesis, and dissertations were included in the study, whereas qualitative study design, single case study research reports, not fully accessed articles, and poor methodological quality were excluded from the analysis.

As suggest in the first revision: “Poor methodological quality were excluded from the analysis". - Poor methodological quality is not a criterion for excluding studies.

After the critical appraisal, the reviewers decided to include or exclude screened articles based on the overall quality of the appraisal score out of 9. The article was prone to exclude when the score was below average, which is of three independent reviewers.

Authors Response: thank you reviewer this is problem write up it is not due methodological quality rather studies due to the low quality appraisal score were excluded in the analysis of the study.

Review reply: Since no studies were excluded because of low quality (which is methodologically inappropriate), I suggest removing this sentence to avoid confusion.

As suggest in the first revision: I recommend using GRADE to assess the quality of evidence generated by meta-analysis.

Authors Response: We had already GRADE and submitted as supplementary file.

See page line. Page 17: line 426

Review reply: GRADE is different to Newcastle-Ottawa Scale or JBI see the reference Guyatt, Gordon H., et al. "GRADE guidelines: a new series of articles in the Journal of Clinical Epidemiology." Journal of Clinical Epidemiology 64.4 (2011): 380-382.

Results

Line 246 – Correct typ”e” one diabetes

Line 252 Description of included studies

“All articles were conducted with a cross-sectional study design with the smallest prevalence from Oromia (11.9%) [34] and the largest prevalence from Amhara 73.6% [35] regional states of Ethiopia. On the contrary, the largest sample size was from the Oromia (1191) [36], whereas the smallest sample size was from Amhara (302) regional states of Ethiopia [37].”

This point must be used in discussion.

Line 263 Quality assessment of included studies

The outcome of the quality appraisal ranged from moderate to high methodological quality, in which two studies scored 9 points [36, 39], three studies scored 8 [34, 35, 38], and the other two studies scored 7 268 [37] [40] (S3 file).

Discussion

As pointed out above, this topic “All articles were conducted with a cross-sectional study design with the smallest prevalence from Oromia (11.9%) [34] and the largest prevalence from Amhara 73.6% [35] regional states of Ethiopia. On the contrary, the largest sample size was from the Oromia (1191) [36], whereas the smallest sample size was from Amhara (302) regional states of Ethiopia [37].” must be used in discussion. For example, the largest Amhara prevalence cannot be influenced by small sample size? In addition, the same point may be made in the study limitations.

Again, I suggest that the quality of the evidence (GRADE) be examined in each of the topics of discussion and in the study limitations.

Conclusion

In the conclusion of the study, my suggestion is to minimize the impact of the results due to the limitations that are presented.

Reviewers' comments:

Reviewer's Responses to Questions

**Comments to the Author**

1. If the authors have adequately addressed your comments raised in a previous round of review and you feel that this manuscript is now acceptable for publication, you may indicate that here to bypass the “Comments to the Author” section, enter your conflict of interest statement in the “Confidential to Editor” section, and submit your "Accept" recommendation.

Reviewer #2: All comments have been addressed

Reviewer #3: (No Response)

2. Is the manuscript technically sound, and do the data support the conclusions?

Reviewer #2: Yes

Reviewer #3: Partly

3. Has the statistical analysis been performed appropriately and rigorously? 

Reviewer #2: Yes

Reviewer #3: Yes

4. Have the authors made all data underlying the findings in their manuscript fully available?

Reviewer #2: Yes

Reviewer #3: No

5. Is the manuscript presented in an intelligible fashion and written in standard English?

Reviewer #2: Yes

Reviewer #3: Yes

6. Review Comments to the Author

Reviewer #2: Thank you for you point-by-point response.

All comments and suggestions have been adequately addressed by the authors.

Reviewer #3: Dear Editor and Authors!

Thank you for the opportunity to review the paper again. In this analysis I can see the effort and progress of the work, but in my opinion, there are still adjustments to be made that could improve the quality of the study.

Abstract – Use the PRISMA for Abstracts Checklist – include PROSPERO registration number

Methods

Searching strategies

Line 118 – Distinguish in the text between searches in traditional databases and searches in gray literature. In addition, Figure 1 needs to be modified to show searches and results from indexed databases and grey literature separately.

In addition, there are other databases besides Google for searching gray literature, clinicaltrials.gov, International Clinical Trials Registry Platform (ICTRP), The European Union Clinical Trials, OpenGrey, ProQuest.

125 – Correct the word "Cochrane" wherever it appears.

130 – S2 file - Present the complete search strategy used in each included database in the Supplementary Appendix.

137 – Correct the word MedLine

149 – Correct the word PubMed

In the text, state that there were no year or language restrictions in the database search.

Eligibility criteria

Study Inclusion and Exclusion criteria

Lines 157 to 160

Quantitative studies that reported the prevalence of overall physical exercise non-adherence of type 2 diabetic patients, master's thesis, and dissertations were included in the study, whereas qualitative study design, single case study research reports, not fully accessed articles, and poor methodological quality were excluded from the analysis.

As suggest in the first revision: “Poor methodological quality were excluded from the analysis". - Poor methodological quality is not a criterion for excluding studies.

After the critical appraisal, the reviewers decided to include or exclude screened articles based on the overall quality of the appraisal score out of 9. The article was prone to exclude when the score was below average, which is of three independent reviewers.

Authors Response: thank you reviewer this is problem write up it is not due methodological quality rather studies due to the low quality appraisal score were excluded in the analysis of the study.

Review reply: Since no studies were excluded because of low quality (which is methodologically inappropriate), I suggest removing this sentence to avoid confusion.

As suggest in the first revision: I recommend using GRADE to assess the quality of evidence generated by meta-analysis.

Authors Response: We had already GRADE and submitted as supplementary file.

See page line. Page 17: line 426

Review reply: GRADE is different to Newcastle-Ottawa Scale or JBI see the reference Guyatt, Gordon H., et al. "GRADE guidelines: a new series of articles in the Journal of Clinical Epidemiology." Journal of Clinical Epidemiology 64.4 (2011): 380-382.

Results

Line 246 – Correct typ”e” one diabetes

Line 252 Description of included studies

“All articles were conducted with a cross-sectional study design with the smallest prevalence from Oromia (11.9%) [34] and the largest prevalence from Amhara 73.6% [35] regional states of Ethiopia. On the contrary, the largest sample size was from the Oromia (1191) [36], whereas the smallest sample size was from Amhara (302) regional states of Ethiopia [37].”

This point must be used in discussion.

Line 263 Quality assessment of included studies

The outcome of the quality appraisal ranged from moderate to high methodological quality, in which two studies scored 9 points [36, 39], three studies scored 8 [34, 35, 38], and the other two studies scored 7 268 [37] [40] (S3 file).

Discussion

As pointed out above, this topic “All articles were conducted with a cross-sectional study design with the smallest prevalence from Oromia (11.9%) [34] and the largest prevalence from Amhara 73.6% [35] regional states of Ethiopia. On the contrary, the largest sample size was from the Oromia (1191) [36], whereas the smallest sample size was from Amhara (302) regional states of Ethiopia [37].” must be used in discussion. For example, the largest Amhara prevalence cannot be influenced by small sample size? In addition, the same point may be made in the study limitations.

Again, I suggest that the quality of the evidence (GRADE) be examined in each of the topics of discussion and in the study limitations.

Conclusion

In the conclusion of the study, my suggestion is to minimize the impact of the results due to the limitations that are presented.

7. PLOS authors have the option to publish the peer review history of their article (what does this mean?). If published, this will include your full peer review and any attached files.

Reviewer #2: **Yes: **Giuseppe Potrick Stefani

Reviewer #3: **Yes: **Luís Fernando Deresz

---

## [Author Response · Author response to Decision Letter 1]

15 Aug 2024

PONE-D-23-20714R1

A systematic review and meta-analysis of physical exercise non-adherence and its determinants among type 2 diabetic patients in Ethiopia:

PLOS ONE

Dear Dr. Abate,

Kind regards,

Thiago Gomes Heck, Ph.D.

Academic Editor

PLOS ONE

Journal Requirements:

Author Response: All reference were checked and completed. 

Additional Editor Comments:

Dear authors,

The manuscript is interestinlgy and can be accept after minor changes.

Please observe carefully the comments made by reviewer 2 below:

Dear Editor and Authors!

Thank you for the opportunity to review the paper again. In this analysis I can see the effort and progress of the work, but in my opinion, there are still adjustments to be made that could improve the quality of the study.

Abstract – Use the PRISMA for Abstracts Checklist – include PROSPERO registration number

Author response: we had used the PRISMA for Abstracts Checklist uploaded in the system and include the ROSPERO online database registration number of CRD42023430579 in method section 

Methods

Searching strategies

Line 118 – Distinguish in the text between searches in traditional databases and searches in gray literature. In addition, Figure 1 needs to be modified to show searches and results from indexed databases and grey literature separately.

Authors’ response: we had distinguished the text searches in traditional databases and searches in gray literature and figure one is modified accordingly. 

In addition, there are other databases besides Google for searching gray literature, clinicaltrials.gov, International Clinical Trials Registry Platform (ICTRP), The European Union Clinical Trials, OpenGrey, ProQuest.

Authors’ response: we made used additional search engine like opengery, proQuest….

125 – Correct the word "Cochrane" wherever it appears.

Authors’ response: Thank you we had made correction on the word “Cochrane”

130 – S2 file - Present the complete search strategy used in each included database in the Supplementary Appendix.

Authors’ response: we had supplied complete searching strategies of the included database. 

137 – Correct the word MedLine

Authors’ response: we had corrected it 

149 – Correct the word PubMed

Authors’ response: we had corrected it

In the text, state that there were no year or language restrictions in the database search.

Authors’ response: Thank you, we had made correction as per your comment. 

Eligibility criteria

Study Inclusion and Exclusion criteria

Lines 157 to 160

Quantitative studies that reported the prevalence of overall physical exercise non-adherence of type 2 diabetic patients, master's thesis, and dissertations were included in the study, whereas qualitative study design, single case study research reports, not fully accessed articles, and poor methodological quality were excluded from the analysis.

As suggest in the first revision: “Poor methodological quality were excluded from the analysis". - Poor methodological quality is not a criterion for excluding studies.

After the critical appraisal, the reviewers decided to include or exclude screened articles based on the overall quality of the appraisal score out of 9. The article was prone to exclude when the score was below average, which is of three independent reviewers.

Authors Response: thank you reviewer this is problem write up it is not due methodological quality rather studies due to the low quality appraisal score were excluded in the analysis of the study.

Review reply: Since no studies were excluded because of low quality (which is methodologically inappropriate), I suggest removing this sentence to avoid confusion.

Authors’ response: Thank you we had removed the text. 

As suggest in the first revision: I recommend using GRADE to assess the quality of evidence generated by meta-analysis.

Authors Response: We had already GRADE and submitted as supplementary file.

See page line. Page 17: line 426

Review reply: GRADE is different to Newcastle-Ottawa Scale or JBI see the reference Guyatt, Gordon H., et al. "GRADE guidelines: a new series of articles in the Journal of Clinical Epidemiology." Journal of Clinical Epidemiology 64.4 (2011): 380-382.

Authors response: we had used the GRADE ON the method section to assess the included study. 

Results

Line 246 – Correct typ”e” one diabetes

Authors response: We had made correction 

Line 252 Description of included studies

“All articles were conducted with a cross-sectional study design with the smallest prevalence from Oromia (11.9%) [34] and the largest prevalence from Amhara 73.6% [35] regional states of Ethiopia. On the contrary, the largest sample size was from the Oromia (1191) [36], whereas the smallest sample size was from Amhara (302) regional states of Ethiopia [37].”

This point must be used in discussion.

Author response: thank you reviewer, this is just for description of the included study it is not the objective of the meta-analysis. We had clearly discussed the prevalence its determinants as the objective of the study. 

Line 263 Quality assessment of included studies

The outcome of the quality appraisal ranged from moderate to high methodological quality, in which two studies scored 9 points [36, 39], three studies scored 8 [34, 35, 38], and the other two studies scored 7 268 [37] [40] (S3 file).

Author response: We had revised it, and we had removed the methodological quality score and changed quality score of the included study based on Newcastle-Ottawa Scale score Discussion

As pointed out above, this topic “All articles were conducted with a cross-sectional study design with the smallest prevalence from Oromia (11.9%) [34] and the largest prevalence from Amhara 73.6% [35] regional states of Ethiopia. On the contrary, the largest sample size was from the Oromia (1191) [36], whereas the smallest sample size was from Amhara (302) regional states of Ethiopia [37].” must be used in discussion. For example, the largest Amhara prevalence cannot be influenced by small sample size? In addition, the same point may be made in the study limitations.

Author response: Thank you, The above information is just description of the included study. The prevalence and the sample size of each study may not be inline since the participants from total study might be different. For example if a study participant is higher in Amhara region compared to Oromia the prevalence from Amhara region becomes higher. This can be checked from each study’s findings. Further, the objective of this study is to find and discussed the pooled prevalence exercise non adherence not the finding of the other researcher individual prevalence. 

Again, I suggest that the quality of the evidence (GRADE) be examined in each of the topics of discussion and in the study limitations.

Author response: We had submitted GRADE quality of evidence in method and result section and limitation section of the study.

Conclusion

In the conclusion of the study, my suggestion is to minimize the impact of the results due to the limitations that are presented.

Author response: we had made revision on it. 

Reviewers' comments:

Reviewer's Responses to Questions

Comments to the Author

1. If the authors have adequately addressed your comments raised in a previous round of review and you feel that this manuscript is now acceptable for publication, you may indicate that here to bypass the “Comments to the Author” section, enter your conflict of interest statement in the “Confidential to Editor” section, and submit your "Accept" recommendation.

Reviewer #2: All comments have been addressed

Reviewer #3: (No Response)

Author: thank you reviewer 2 for your constructive comments

2. Is the manuscript technically sound, and do the data support the conclusions?

Reviewer #2: Yes

Reviewer #3: Partly

We had made revision for reviewer 3

3. Has the statistical analysis been performed appropriately and rigorously?

Reviewer #2: Yes

Reviewer #3: Yes

Author response thank you reviewers 

4. Have the authors made all data underlying the findings in their manuscript fully available?

Reviewer #2: Yes

Reviewer #3: No

Author Response: thank you reviwers we had add Data Availability Statement and other requirement of the plos one journal

5. Is the manuscript presented in an intelligible fashion and written in standard English?

Reviewer #2: Yes

Reviewer #3: Yes

Author Response: thank you reviewers for your constructive comments 

6. Review Comments to the Author

Reviewer #2: Thank you for you point-by-point response.

All comments and suggestions have been adequately addressed by the authors.

Reviewer #3: Dear Editor and Authors!

Thank you for the opportunity to review the paper again. In this analysis I can see the effort and progress of the work, but in my opinion, there are still adjustments to be made that could improve the quality of the study.

Abstract – Use the PRISMA for Abstracts Checklist – include PROSPERO registration number

Author response: we had used the PRISMA for Abstracts Checklist uploaded in the system and include the ROSPERO online database registration number of CRD42023430579 in method section

Methods

Searching strategies

Line 118 – Distinguish in the text between searches in traditional databases and searches in gray literature. In addition, Figure 1 needs to be modified to show searches and results from indexed databases and grey literature separately.

Authors’ response: we had distinguished the text searches in traditional databases and searches in gray literature and figure one is modified accordingly. 

In addition, there are other databases besides Google for searching gray literature, clinicaltrials.gov, International Clinical Trials Registry Platform (ICTRP), The European Union Clinical Trials, OpenGrey, ProQuest.

Authors’ response: we made used additional search engine like opengery, proQuest….

125 – Correct the word "Cochrane" wherever it appears.

Authors’ response: Thank you we had made correction on the word “Cochrane”

130 – S2 file - Present the complete search strategy used in each included database in the Supplementary Appendix.

Authors’ response: we had supplied complete searching strategies of the included database

137 – Correct the word MedLine

Authors’ response: we had corrected it

149 – Correct the word PubMed

Authors’ response: we had corrected it 

In the text, state that there were no year or language restrictions in the database search.

Authors’ response: Thank you, we had made correction as per your comment 

Eligibility criteria

Study Inclusion and Exclusion criteria

Lines 157 to 160

Quantitative studies that reported the prevalence of overall physical exercise non-adherence of type 2 diabetic patients, master's thesis, and dissertations were included in the study, whereas qualitative study design, single case study research reports, not fully accessed articles, and poor methodological quality were excluded from the analysis.

As suggest in the first revision: “Poor methodological quality were excluded from the analysis". - Poor methodological quality is not a criterion for excluding studies.

After the critical appraisal, the reviewers decided to include or exclude screened articles based on the overall quality of the appraisal score out of 9. The article was prone to exclude when the score was below average, which is of three independent reviewers.

Authors Response: thank you reviewer this is problem write up it is not due methodological quality rather studies due to the low quality appraisal score were excluded in the analysis of the study.

Review reply: Since no studies were excluded because of low quality (which is methodologically inappropriate), I suggest removing this sentence to avoid confusion.

Authors’ response: Thank you we had removed the text. 

As suggest in the first revision: I recommend using GRADE to assess the quality of evidence generated by meta-analysis.

Authors Response: We had already GRADE and submitted as supplementary file.

See page line. Page 17: line 426

Review reply: GRADE is different to Newcastle-Ottawa Scale or JBI see the reference Guyatt, Gordon H., et al. "GRADE guidelines: a new series of articles in the Journal of Clinical Epidemiology." Journal of Clinical Epidemiology 64.4 (2011): 380-382.

Authors response: we had used the GRADE ON the method section to assess the included study.

Results

Line 246 – Correct typ”e” one diabetes

Authors response: We had made correction

Line 252 Description of included studies

“All articles were conducted with a cross-sectional study design with the smallest prevalence from Oromia (11.9%) [34] and the largest prevalence from Amhara 73.6% [35] regional states of Ethiopia. On the contrary, the largest sample size was from the Oromia (1191) [36], whereas the smallest sample size was from Amhara (302) regional states of Ethiopia [37].”

This point must be used in discussion.

Line 263 Quality assessment of included studies

The outcome of the quality appraisal ranged from moderate to high methodological quality, in which two studies scored 9 points [36, 39], three studies scored 8 [34, 35, 38], and the other two studies scored 7 268 [37] [40] (S3 file).

Author response: thank you reviewer, this is just for description of the included study it is not the objective of the meta-analysis. We had clearly discussed the prevalence its determinants as the objective of the study. 

Discussion

As pointed out above, this topic “All articles were conducted with a cross-sectional study design with the smallest prevalence from Oromia (11.9%) [34] and the largest prevalence from Amhara 73.6% [35] regional states of Ethiopia. On the contrary, the largest sample size was from the Oromia (1191) [36], whereas the smallest sample size was from Amhara (302) regional states of Ethiopia [37].” must be used in discussion. For example, the largest Amhara preva

---

## [Decision Letter · Decision Letter 2]

15 Oct 2024

PONE-D-23-20714R2A systematic review and meta-analysis of physical exercise non-adherence and its determinants among type 2 diabetic patients in Ethiopia:PLOS ONE

Dear Dr. Abate,

Thank you for submitting your manuscript to PLOS ONE. After careful consideration, we feel that it has merit but does not fully meet PLOS ONE’s publication criteria as it currently stands. Therefore, we invite you to submit a revised version of the manuscript that addresses the points raised during the review process.

We look forward to receiving your revised manuscript.

Kind regards,

Philipp Baumert

Academic Editor

PLOS ONE

Journal Requirements:

Additional Editor Comments:

Dear Hailemichael Kindie Abate and Co-authors,

Thank you for your thorough revision of the manuscript. I was recently assigned the role of academic editor and generally aim to minimise the need for repeated revisions. However, due to my later involvement in this process, I encourage you to carefully address the comments from Reviewer 2. With these adjustments, I am confident that your manuscript will be strengthened, and I hope we can finalise this process promptly.

To clarify a few points:

- Line 120: As noted by Reviewer 2, Google Scholar is not a traditional indexed database. Please include it under grey literature and add the missing comma as suggested.

- Figure 1: Am I right that the search results for OpenGrey and ProQuest are missing in figure 1? Please kindly include these.

- Lines 169–171: Reviewer 2 has suggested removing this sentence if no articles were removed based on quality checks. Please either provide an example of excluded articles in your revision or delete this sentence if none were removed.

- Lastly, in the conclusion (full text and Abstract), please consider adding a brief statement that the results should be interpreted with caution due to the low certainty of evidence. You may choose to reference the GRADE score in this context (or not).

Thank you for your attention to these details, and I look forward to receiving the revised manuscript.

Best wishes,

Philipp Baumert

Reviewers' comments:

Reviewer's Responses to Questions

**Comments to the Author**

1. If the authors have adequately addressed your comments raised in a previous round of review and you feel that this manuscript is now acceptable for publication, you may indicate that here to bypass the “Comments to the Author” section, enter your conflict of interest statement in the “Confidential to Editor” section, and submit your "Accept" recommendation.

Reviewer #2: All comments have been addressed

Reviewer #3: (No Response)

2. Is the manuscript technically sound, and do the data support the conclusions?

Reviewer #2: Yes

Reviewer #3: Partly

3. Has the statistical analysis been performed appropriately and rigorously? 

Reviewer #2: Yes

Reviewer #3: Yes

4. Have the authors made all data underlying the findings in their manuscript fully available?

Reviewer #2: Yes

Reviewer #3: No

5. Is the manuscript presented in an intelligible fashion and written in standard English?

Reviewer #2: Yes

Reviewer #3: Yes

6. Review Comments to the Author

Reviewer #2: Thank you for you point-by-point response.

All comments and suggestions have been adequately addressed by the authors.

Reviewer #3: Dear Editor and Authors!

Thank you for the opportunity to review the paper again.

I believe that the use of GRADE has added robustness to the results obtained.

Here are a few suggestions to include in the paper.

118 Searching strategies

119 The indexed traditional database (Medline/PubMed, Scopus, Web of Science, and Cochrane

120 Library Google Scholar) – Google Scholar isn’t an indexed database.

132 strategy design for Embase Cochrane – put coma between Embase and Cochrane

137 The first search through PubMed, Cochran Library – please, correct Cochran”e”

161 Quality appraisal of included studies

162 The articles searched in the database were collected and duplicate articles were removed

163 manually using EndNote (version 7). This sentence can be deleted – is duplicate – line 151 and 152.

169 ... After the critical appraisal, the reviewers decided to include or exclude screened articles

170 based on the overall quality of the appraisal score out of 9. The article was prone to exclude

171 when the score was below average, which is of three independent reviewers. –

Authors Response (R1): thank you reviewer this is problem write up it is not due methodological quality rather studies due to the low quality appraisal score were excluded in the analysis of the study.

Review reply: Since no studies were excluded because of low quality (which is methodologically inappropriate), I suggest removing this sentence to avoid confusion.

Review reply 2 – To avoid confusion, I strongly recommend removing this sentence.

Additional suggestions:

Since the authors used GRADE, I suggest including it in the abstract (methods and results). In addition, I suggest rewording the conclusion (of the article and the abstract) to emphasize that the results should be interpreted with caution due to the low certainty of evidence indicated by GRADE.

7. PLOS authors have the option to publish the peer review history of their article (what does this mean?). If published, this will include your full peer review and any attached files.

Reviewer #2: **Yes: **Giuseppe Potrick Stefani

Reviewer #3: No

---

## [Author Response · Author response to Decision Letter 2]

1 Nov 2024

Editorial office comments and response 

We look forward to receiving your revised manuscript.

Kind regards,

Philipp Baumert

Academic Editor

PLOS ONE

Journal Requirements:

Author Response: All reference were checked and completed.

Additional Editor Comments:

Dear Hailemichael Kindie Abate and Co-authors,

Thank you for your thorough revision of the manuscript. I was recently assigned the role of academic editor and generally aim to minimise the need for repeated revisions. However, due to my later involvement in this process, I encourage you to carefully address the comments from Reviewer 2. With these adjustments, I am confident that your manuscript will be strengthened, and I hope we can finalise this process promptly.

To clarify a few points:

- Line 120: As noted by Reviewer 2, Google Scholar is not a traditional indexed database. Please include it under grey literature and add the missing comma as suggested.

Author Response; Thank you we had made correction as per your comment. 

- Figure 1: Am I right that the search results for OpenGrey and ProQuest are missing in figure 1? Please kindly include these.

Author Response: Thank you, we had made correction on figure 1

- Lines 169–171: Reviewer 2 has suggested removing this sentence if no articles were removed based on quality checks. Please either provide an example of excluded articles in your revision or delete this sentence if none were removed.

Author Response: thank you we had cited the articles removed from the analysis based on quality checks 

See page 7 Line 178 and the supplementary file for quality checks 

- - Lastly, in the conclusion (full text and Abstract), please consider adding a brief statement that the results should be interpreted with caution due to the low certainty of evidence. You may choose to reference the GRADE score in this context (or not).

Author Response: Whe had put the result with text and supplementary file. See page 9 line 223-2232 page: 14 line 336-351 the detail on result section. We also stated on the abstract section about the GRADE. 

Thank you for your attention to these details, and I look forward to receiving the revised manuscript.

Best wishes,

Philipp Baumert

Reviewers' comments:

Reviewer's Responses to Questions

Comments to the Author

1. If the authors have adequately addressed your comments raised in a previous round of review and you feel that this manuscript is now acceptable for publication, you may indicate that here to bypass the “Comments to the Author” section, enter your conflict of interest statement in the “Confidential to Editor” section, and submit your "Accept" recommendation.

Reviewer #2: All comments have been addressed

Reviewer #3: (No Response)

2. Is the manuscript technically sound, and do the data support the conclusions?

Reviewer #2: Yes

Reviewer #3: Partly

Author response: We had described the result and concluded in supportive manner 

3. Has the statistical analysis been performed appropriately and rigorously?

Reviewer #2: Yes

Reviewer #3: Yes

4. Have the authors made all data underlying the findings in their manuscript fully available?

Reviewer #2: Yes

Reviewer #3: No

Author response: all data is described in the manuscript and supported with supplementary data in the revised manuscript 

5. Is the manuscript presented in an intelligible fashion and written in standard English?

Reviewer #2: Yes

Reviewer #3: Yes

6. Review Comments to the Author

Reviewer #2: Thank you for you point-by-point response.

All comments and suggestions have been adequately addressed by the authors.

Reviewer #3: Dear Editor and Authors!

Thank you for the opportunity to review the paper again.

I believe that the use of GRADE has added robustness to the results obtained.

Here are a few suggestions to include in the paper.

118 Searching strategies

Author response: we had all the comments under searching strategies. See the searching strategies section.

119 The indexed traditional database (Medline/PubMed, Scopus, Web of Science, and Cochrane

Author response: We had corrected as per your comments 

120 Library Google Scholar) – Google Scholar isn’t an indexed database.

Author response: thank you we had corrected under grey literature data base

132 strategy design for Embase Cochrane – put coma between Embase and Cochrane

Author response: Thank you we had corrected comma 

137 The first search through PubMed, Cochran Library – please, correct Cochran”e”

Author response: Thank you we had corrected

161 Quality appraisal of included studies

162 The articles searched in the database were collected and duplicate articles were removed

163 manually using EndNote (version 7). This sentence can be deleted – is duplicate – line 151 and 152.

Author response: We had removed the text as per your comment.

169 ... After the critical appraisal, the reviewers decided to include or exclude screened articles

170 based on the overall quality of the appraisal score out of 9. The article was prone to exclude

171 when the score was below average, which is of three independent reviewers. –

Authors Response (R1): thank you reviewer this is problem write up it is not due methodological quality rather studies due to the low quality appraisal score were excluded in the analysis of the study.

Review reply: Since no studies were excluded because of low quality (which is methodologically inappropriate), I suggest removing this sentence to avoid confusion.

Review reply 2 – To avoid confusion, I strongly recommend removing this sentence.

Author Response: thank you we had cited the articles removed from the analysis based on quality checks See page 7 Line 178 and the supplementary file for quality checks

Additional suggestions:

Since the authors used GRADE, I suggest including it in the abstract (methods and results). In addition, I suggest rewording the conclusion (of the article and the abstract) to emphasize that the results should be interpreted with caution due to the low certainty of evidence indicated by GRADE.

Author Response: Whe had put the result with text and supplementary file. See page 9 line 223-2232 page: 14 line 336-351 the detail on result section. We also stated on the abstract section about the GRADE. 

7. PLOS authors have the option to publish the peer review history of their article (what does this mean?). If published, this will include your full peer review and any attached files.

Do you want your identity to be public for this peer review? For information about this choice, including consent withdrawal, please see our Privacy Policy.

Reviewer #2: Yes: Giuseppe Potrick Stefani

Reviewer #3: No

While revising your submission, please upload your figure files to the Preflight Analysis and Conversion Engine (PACE) digital diagnostic tool, https://pacev2.apexcovantage.com/. PACE helps ensure that figures meet PLOS requirements. To use PACE, you must first register as a user. Registration is free. Then, login and navigate to the UPLOAD tab, where you will find detailed instructions on how to use the tool. If you encounter any issues or have any questions when using PACE, please email PLOS at <a href="mailto:figures@plos.org">figures@plos.org. Please note that Supporting Information files do not need this step.

---

## [Editor Report · Decision Letter 3]

11 Nov 2024

A systematic review and meta-analysis of physical exercise non-adherence and its determinants among type 2 diabetic patients in Ethiopia:

PONE-D-23-20714R3

Dear Dr. Abate,

We’re pleased to inform you that your manuscript has been judged scientifically suitable for publication and will be formally accepted for publication once it meets all outstanding technical requirements.

Kind regards,

Philipp Baumert

Academic Editor

PLOS ONE

---

## [Editor Report · Acceptance letter]

21 Nov 2024

PONE-D-23-20714R3 

PLOS ONE

Dear Dr. Abate, 

I'm pleased to inform you that your manuscript has been deemed suitable for publication in PLOS ONE. Congratulations! Your manuscript is now being handed over to our production team.

Kind regards, 

on behalf of

Dr. Philipp Baumert 

Academic Editor

PLOS ONE